# An unexpected role for the conserved ADAM-family metalloprotease ADM-2 in *Caenorhabditis elegans* molting

**Braveen B. Joseph**[ORCID]**, Phillip T. Edeen**[ORCID]**, Sarina Meadows**[ORCID]**, Shaonil Binti**[ORCID]**, David S. Fay**[ORCID]*

Department of Molecular Biology, College of Agriculture and Natural Resources, University of Wyoming, Laramie, Wyoming, United States of America

* davidfay@uwyo.edu

**Data Availability Statement:** All raw data is included in S1 File.

**Funding:** This project was supported by R35 GM136236 to DSF and by an Institutional

## Abstract

Molting is a widespread developmental process in which the external extracellular matrix (ECM), the cuticle, is remodeled to allow for organismal growth and environmental adaptation. Studies in the nematode *Caenorhabditis elegans* have identified a diverse set of molting-associated factors including signaling molecules, intracellular trafficking regulators, ECM components, and ECM-modifying enzymes such as matrix metalloproteases. *C. elegans* NEKL-2 and NEKL-3, two conserved members of the NEK family of protein kinases, are essential for molting and promote the endocytosis of environmental steroid-hormone precursors by the epidermis. Steroids in turn drive the cyclic induction of many genes required for molting. Here we report a role for the sole *C. elegans* ADAM–meltrin metalloprotease family member, ADM-2, as a mediator of molting. Loss of *adm-2*, including mutations that disrupt the metalloprotease domain, led to the strong suppression of molting defects in partial loss-of-function *nekl* mutants. ADM-2 is expressed in the epidermis, and its trafficking through the endo-lysosomal network was disrupted after NEKL depletion. We identified the epidermally expressed low-density lipoprotein receptor–related protein, LRP-1, as a candidate target of ADM-2 regulation. Whereas loss of ADM-2 activity led to the upregulation of apical epidermal LRP-1, ADM-2 overexpression caused a reduction in LRP-1 levels. Consistent with this, several mammalian ADAMs, including the meltrin ADAM12, have been shown to regulate mammalian LRP1 via proteolysis. In contrast to mammalian homologs, however, the regulation of LRP-1 by ADM-2 does not appear to involve the metalloprotease function of ADM-2, nor is proteolytic processing of LRP-1 strongly affected in *adm-2* mutants. Our findings suggest a noncanonical role for an ADAM family member in the regulation of a lipoprotein-like receptor and lead us to propose that endocytic trafficking may be important for both the internalization of factors that promote molting as well as the removal of proteins that can inhibit the process.

Development Award (IDeA) from the National Institute of General Medical Sciences of the National Institutes of Health (P20GM103432) to the University of Wyoming. The funders had no role in study design, data collection and analysis, decision to publish, or preparation of the manuscript. https://www.nih.gov.

**Competing interests:** The authors have declared that no competing interests exist.

## Author summary

The molecular and cellular features of molting in nematodes share many similarities with cellular and developmental processes that occur in mammals. This includes the degradation and reorganization of extracellular matrix materials that surround cells, as well as the intracellular machineries that allow cells to sample and modify their environments. In the current study, we found an unexpected function for a conserved protein that cleaves other proteins on the external surface of cells. Rather than promoting molting through extracellular matrix reorganization, however, the ADM-2 protease appears to function as a negative regulator of molting. This observation can be explained in part by data showing that ADM-2 negatively regulates a cell surface receptor required for molting. Surprisingly, it appears to do so through a mechanism that does not involve proteolysis. Our data provide insights into the mechanisms controlling molting and link several conserved proteins to show how they function together during development.

## Introduction

The cuticle of *Caenorhabditis elegans* is an external extracellular matrix (ECM) required for locomotion, body shape maintenance, and protection from the environment [1,2]. During larval development *C. elegans* undergoes four molts, a specialized form of apical ECM remodeling, whereby a new cuticle is synthesized under the old cuticle, which is partially degraded and shed [1,2]. *C. elegans* molting cycles are orchestrated by conserved steroid-hormone receptors, including NHR-23 (an ortholog of human RORC) and NHR-25 (an ortholog of human NR5A1), which collectively control the oscillation of hundreds of mRNAs [2–4]. The production of molting-specific steroid-hormone ligands is thought to be dependent on the uptake of environmental sterols by epidermally expressed LRP-1 (the homolog of human LRP2/megalin), a member of the low-density lipoprotein receptor–related protein family [5,6]. Consistent with this, internalization of LRP-1 by clathrin-mediated endocytosis (CME) is essential for normal molting [2,5–7].

We have previously shown that the *C. elegans* protein kinases NEKL-2 (an ortholog of human NEK8/9) and NEKL-3 (an ortholog of human NEK6/7) promote endocytosis of LRP-1 at the apical epidermal plasma membrane. Correspondingly, loss of either NEKL-2 or NEKL-3 function leads to a reduction or delay in the expression of molting genes, a failure to complete molting, and larval arrest [2,8–12]. NEKL-2 and NEKL-3 (NEKLs) are members of the NIMA-related kinase (NEK) protein family, mammalian orthologs of which have been studied primarily in the context of cell cycle regulation and ciliogenesis [13–28]. *C. elegans* NEKLs bind to and co-localize with several conserved ankyrin-repeat proteins including, MLT-2 (an ortholog of human ANKS6), MLT-3 (an ortholog of human ANKS3), and MLT-4 (an ortholog of human INVS), referred to here collectively as MLTs, which are essential for the proper localization of NEKLs [9]. Correspondingly, loss of MLT functions leads to molting defects that are identical to those observed with loss of the NEKLs [9]. NEKLs and MLTs form two distinct complexes (NEKL-2–MLT-2–MLT-4 and NEKL-3–MLT-3) and are expressed in a punctate pattern in the large epidermal syncytium known as hyp7, in which they are specifically required [2,8,9].

The cellular and physiological mechanisms by which NEKLs–MLTs impact the molting process through intracellular trafficking have yet to be fully explored. Previous work suggests that NEKLs are required for the uptake and processing of membrane cargo, including LRP-1, which in turn promote molting [11, 29]. Using a forward-genetics suppressor approach [30],

we previously found that loss of AP2 clathrin-adapter subunits, as well as the AP2 allosteric activator FCHO-1 can suppress molting and trafficking defects in NEKL mutants [11]. These and other studies revealed that NEKLs control endocytosis in part by facilitating the uncoating of sub-apical clathrin-coated vesicles and may affect additional trafficking processes through the regulation of actin via the CDC-42–SID-3 (corresponding to the human CDC42–ACK1/2) pathway [10,11].

Here we report suppression of *nekl* molting defects by loss of the conserved ADM-2 trans-membrane metalloprotease. Although proteases have previously been implicated as positive regulators of molting, our data is consistent with ADM-2 exerting a negative influence on the molting process. ADM-2 belongs to the ADAM (a disintegrin and metalloprotease) family of metallopeptidases, which are members of the zinc protease superfamily [31,32]. ADM-2 is the sole member of the meltrin metalloprotease subfamily in *C. elegans* [33,34], which in humans consists of ADAM9 (Meltrin γ), ADAM12 (Meltrin α), ADAM19 (Meltrin β), and ADAM33 [35,36]. ADAM/meltrin family proteins function as "sheddases", cleaving target peptides that are positioned at or near the outer leaflet of the plasma membrane [31,32,37–51]. Notably, knockouts of meltrin family members in mammals have generally not provided clear insights into the roles of meltrins in vivo during development, which may in part be due to genetic redundancies [33,52,53]. Here we show that, unlike AP2 and *fcho-1* mutants, loss of ADM-2 function did not suppress *nekl*-associated trafficking defects. Rather, ADM-2 was itself dependent on NEKL–MLTs for its trafficking through the endocytic pathway. Moreover, ADM-2 appears to regulate LRP-1 through a mechanism that is independent of its proteolytic activity. Collectively, our studies suggest that NEKLs may be required for the internalization of both positive and negative regulators of molting.

## Results

### *nekl* molting defects are suppressed by reduced function of the ADM-2 metalloprotease

We previously described an approach to identify genetic suppressors of partial loss-of-function mutations in NEKL kinases [30]. Whereas strains homozygous for either *nekl-2(fd81)* or *nekl-3(gk894345)* weak loss-of-function alleles are viable, *nekl-2(fd81)*; *nekl-3(gk894345)* double mutants (hereafter referred to as *nekl-2; nekl-3* mutants) are synthetically lethal and exhibit ~98.5% larval arrest due to L2/L3 molting defects [9]. In the absence of a suppressor mutation, propagation of *nekl-2; nekl-3* mutants requires the presence of a *nekl-2*⁺ or *nekl-3*⁺ transgenic rescuing array (GFP⁺; Fig 1A and 1E). In contrast, strains homozygous for the suppressor alleles *fd130* or *fd163* of are ~80% viable and propagate in the absence of a rescuing array (GFP⁻; Fig 1B, 1C and 1E).

Using our protocols for whole-genome sequencing and bioinformatical analysis [30], we identified the causal mutation corresponding to *fd130* to be a G-to-A transition in exon 10 of *adm-2*/C04A11.4 (Fig 1D). *fd130* converts codon 494 (TGG; W) into a premature translational termination signal (TGA; stop codon), resulting in the predicted truncation of the 952-amino-acid protein. Correspondingly, the independently isolated allele *fd163* is a G-to-A transition in the conserved 5' splice donor sequence in the first intron of *adm-2* (GT to AT) (Fig 1D). The resulting intron-retaining mutation is predicted to result in a stop codon immediately following R66. Using CRISPR/Cas9 methods we isolated several additional *adm-2* alleles including *fd208*, a 1-bp deletion that causes a frameshift after Y479, along with *fd229*, a deletion that spans exons 4–10 and results in frame shifts after S123 (Fig 1D). Like *fd130* and *fd163*, these alleles led to similarly robust suppression of molting defects in the *nekl-2; nekl-3* background and are predicted to result in strong or complete loss of ADM-2 function (Fig 1E).

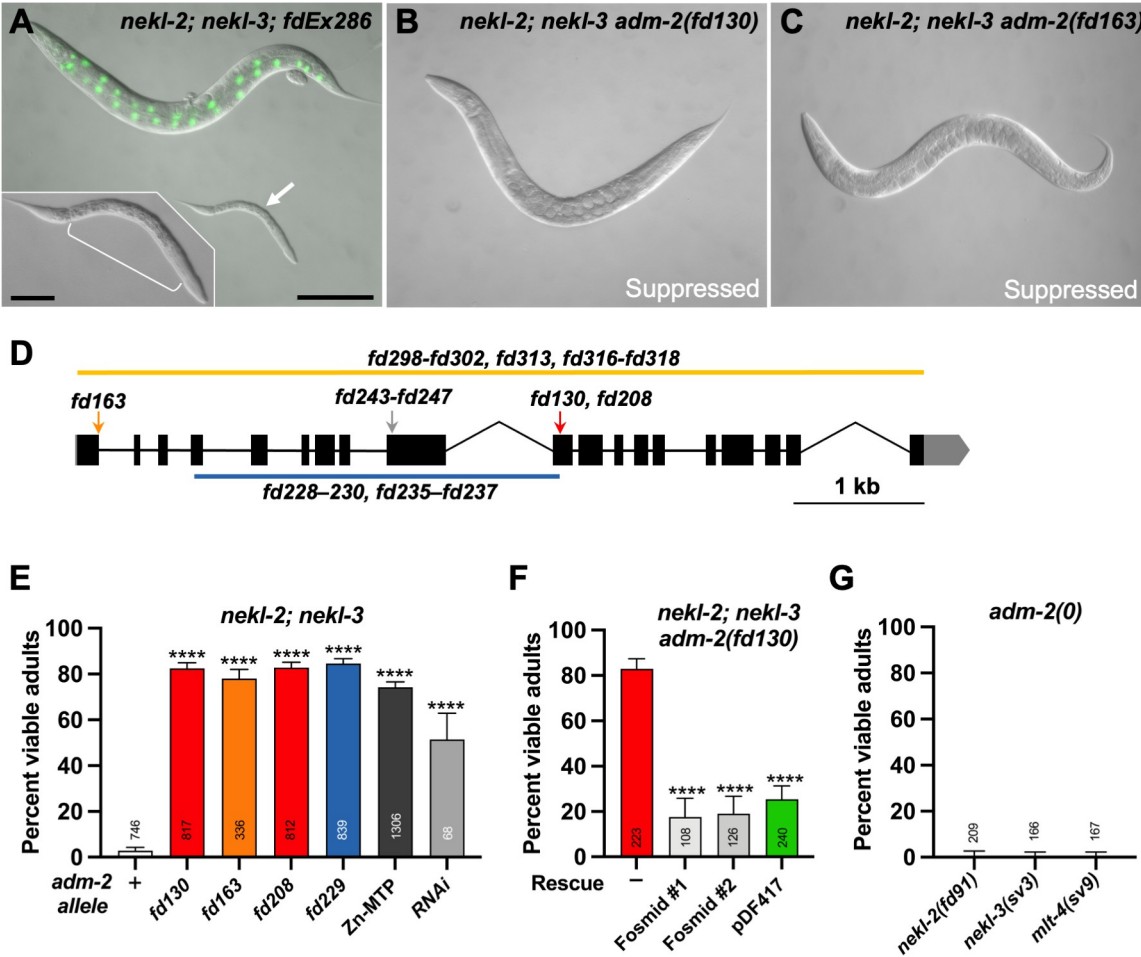

**Fig 1. Loss of ADM-2 function suppresses *nekl* molting defects.** (A) DIC image of *nekl-2; nekl-3* double-mutant worms. The adult worm contains a rescuing extrachromosomal array *(nekl-3⁺; sur-5::GFP)*. An arrested larva is marked by the white arrow and enlarged in the inset; the white bracket indicates the constricted region containing a double cuticle. (B, C) DIC images of *nekl-2; nekl-3* double-mutant adult worms containing the *adm-2(fd130)* (B) and *adm-2(fd163)* (C) mutant alleles. Bar size in A = 100 μm (for A–C); in inset, 20 μm. (D) Schematic diagram of the *adm-2* locus. Solid black rectangles indicate exons; introns are demarcated by black lines. Locations of the *fd163*, *fd130*, *fd208*, and *fd243–fd247* alleles are indicated by arrows. Large deletion alleles *fd298–fd302, fd313, fd316–fd318, fd228–fd230*, and *fd235–fd237* are indicated by orange and blue lines. (E) Bar plot showing percentage of viable adult-stage *nekl-2; nekl-3* worms with the indicated *adm-2* alleles (or RNAi); + indicates wild-type *adm-2*. (F) Bar plot showing reversion of suppression in *nekl-2; nekl-3 adm-2(fd130)* mutants by fosmids expressing wild-type *adm-2* and by an *adm-2* cDNA fused to GFP (pDF417). Fosmid #1 and #2 indicate two independent extrachromosomal lines. (G) Bar plot showing failure to suppress molting defects in *nekl–mlt* hypomorphic mutants by *adm-2* null mutants [*nekl-2(fd91); adm-2(fd313), nekl-3(sv3); adm-2(fd316)*, and *mlt-4(sv9); adm-2(fd317)*]. Error bars in E–G represent 95% confidence intervals. p-Values were determined using Fisher's exact test; ****p ≤ 0.0001. Raw data for this figure is provided in S1 File.

Several additional pieces of evidence indicate that it is loss of ADM-2 function that leads to suppression of *nekl-2; nekl-3* molting defects. (1) *adm-2* mutations that suppress these molting defects (e.g., *fd130* and *fd163*) are fully recessive (see Materials and Methods). (2) Extrachromosomal expression of a fosmid clone containing wild-type genomic ADM-2 sequences strongly reversed suppression in *nekl-2; nekl-3 adm-2* mutants as did a plasmid (pDF417) encoding an ADM-2::GFP fusion under the control of the *adm-2* promoter (Fig 1F). (3) RNAi of *adm-2* led to significant suppression of molting defects in *nekl-2; nekl-3* mutants (Fig 1E). We note that loss of *adm-2* in wild-type backgrounds, including a strong loss-of-function deletion allele, *fd300*, did not appear to impair development, health, or fecundity, indicating that

*adm-2* is a non-essential gene (S1A Fig). Consistent with this, no phenotypes have been previously ascribed to *adm-2* mutations.

ADM-2 is a member of the ADAM (a disintegrin and metalloprotease) family of metallopeptidases, with its closest human homologs belonging to the meltrin subfamily (ADAM9/12/19/33) [33,35,36]. Meltrins are notable for having functional proteases that contain a histidine-coordinated zinc-binding site (HExxHxxGxxH), which is also found in ADM-2 (S2 Fig) [54,55]. To determine if the putative metalloprotease activity of ADM-2 is important for its influence on molting, we CRISPR-engineered an ADM-2 variant in which the three conserved histidine residues within the predicted Zn-binding domain were altered [Zn-MTP: H312–H322 (**H**ELG**H**TF**G**MD**H** > **D**ALAY TF**R**MDY)]. Notably, this change led to the strong suppression of molting defects in *nekl-2; nekl-3* mutants, indicating that the ADM-2 metalloproteinase activity can impact the molting process (Fig 1E). Like other meltrins, ADM-2 also contains an N-terminal cysteine switch, cysteine loop, and disintegrin domain, a transmembrane domain, and several predicted SH3-binding sites in its cytoplasmic C terminus (S2 Fig; also see below) [32,56]. Although linked to a range of human diseases, individual loss-of-function mutations in mouse meltrins have generally not produced robust developmental defects, and no phenotypes have been associated with either of the two *Drosophila* meltrin family members [33,52,53].

We note that WormBase annotates two *adm-2* isoforms that are identical through exon 18 (corresponding to aa A915) but differ at the 5' ends of their 19th (terminal) exons; *adm-2a* and *adm-2b* are predicted to encode 952 and 929 aa proteins, respectively. The noncanonical 18th intron acceptor site of *adm-2b* (5-'GCAAAAG-3') occurs 7 bp upstream of the corresponding acceptor site of *adm-2a* (5'-ATTTCAG-3') and terminates translation 76 bp upstream of the stop codon of *adm-2a*. The peptide regions corresponding to exon 19 of *adm-2a* (37 aa) and *adm-2b* (14 aa) do not contain any known domains nor were homologies detected with other proteins.

To determine if other *C. elegans* ADAM and ADAM-TS family members may also contribute to molting control, we tested ten other family members for their ability to suppress *nekl-2; nekl-3* molting defects (S1B and S1C Fig). We failed to detect suppression after inhibition of each gene using RNAi (dsRNA) injection methods, which were effective in promoting suppression when targeting *adm-2*. Thus, the suppression of *nekl*-associated molting defects by *adm-2* appears unique among the *C. elegans* ADAM family members.

## ADM-2 is expressed in multiple tissues including the epidermis

To gain insight into how ADM-2 may affect molting in *nekl* mutants, we examined endogenously tagged *adm-2*::*mScarlet* and *adm-2*::*GFP* strains, in which the fluorescent marker was fused to the C terminus of the ADM-2a cytoplasmic domain. Both CRISPR-tagged versions showed a punctate pattern within hyp7, a large epidermal syncytium that encompasses most of the central body region of the worm, including localization to small puncta near the apical membrane (Fig 2A–2E'). Notably, both NEKLs and MLTs are specifically expressed and required in the hyp7 syncytium [8,9]. We also detected some differences between the localization patterns of ADM-2::mScarlet and ADM-2::GFP. In particular, ADM-2::mScarlet was observed in larger vesicular and tubular-like structures throughout the epidermis, whereas these structures were mostly absent in ADM-2::GFP worms (Fig 2A–2D; also see below).

ADM-2 was also detectable in seam cells, a lateral epidermal syncytium that borders hyp7 along the length of the animal (Fig 2E and 2E'); in the anterior epidermis (S3A and A'); and in a variety of head, tail, and centrally located neurons (Figs 2E and 2E'; S3B–S3C and S3B'–S3C'). In addition, ADM-2 was observed in proximal oogenic cells of the hermaphrodite

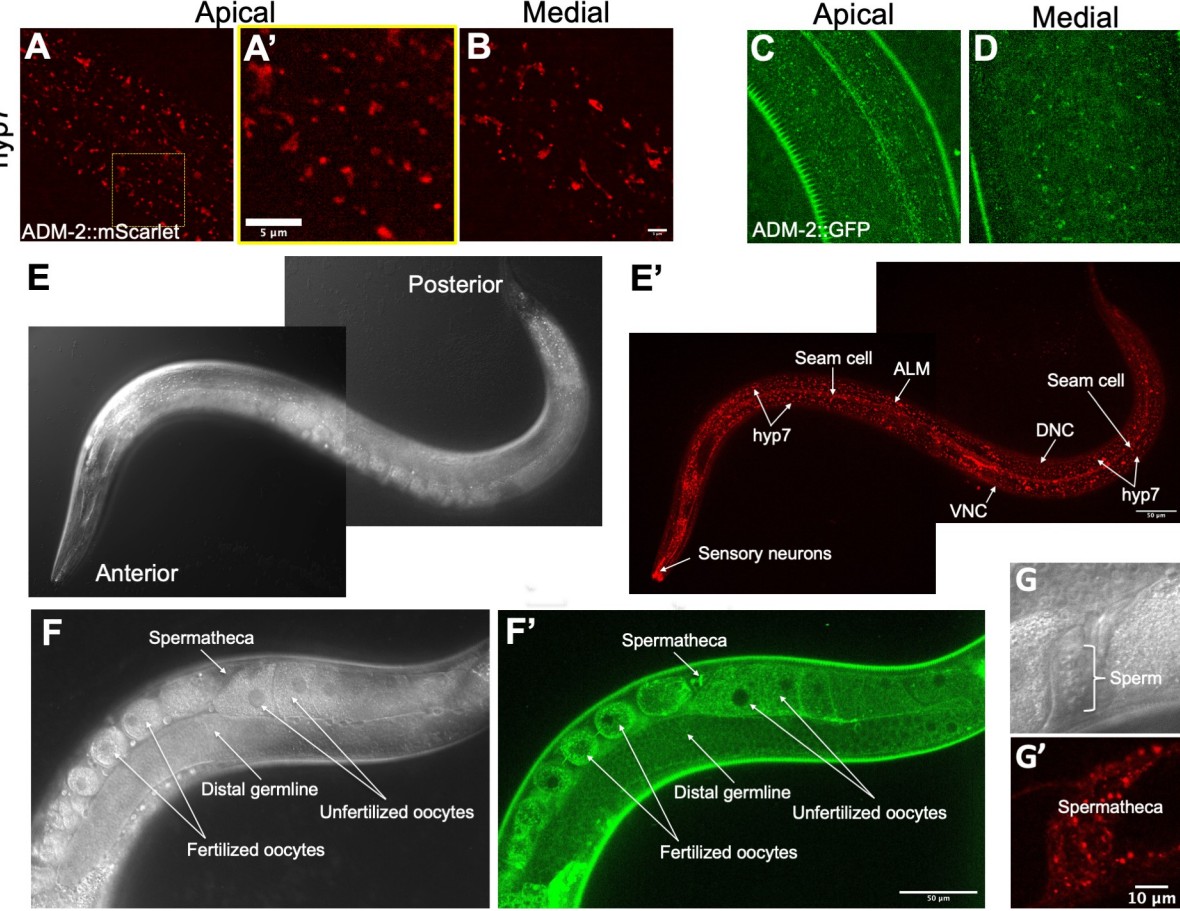

**Fig 2. Expression of endogenously tagged ADM-2.** (A–D and A') Representative confocal images of ADM-2 expression in the *C. elegans* hyp7 region. ADM-2::mScarlet (A, B) and ADM-2::GFP (C, D) expression at apical (A, C) and medial (B, D) planes. A' is the inset from panel A. Bar in B = 5 μm (for A–D); in inset A' = 5 μm. (E–G and E'–G') Representative DIC (E–G) and confocal (E'–G') images of ADM-2 expression. ADM-2::mScarlet in the hyp7 hypodermis; seam cells; sensory neurons; and ALM, DNC, and VNC neurons (E, E') and in spermatheca (G, G'). ADM-2::GFP expression in the distal germline, oocytes, and spermatheca (F, F'). Bar in E' = 50 μm (for E, E'); in G' = 10 μm (for G, G'); in F' = 50 μm (for F, F').

germline, with levels increasing in maturing oocytes, where it was localized to the cytoplasm and plasma membrane (Fig 2F and 2F'). Likewise, ADM-2 is expressed in fertilized oocytes (Fig 2F and 2F') and throughout embryogenesis (S3D–S3F and S3D'–S3F' Fig). In contrast, ADM-2 was not detected in mature sperm cells but was expressed in myoepithelial cells of the hermaphrodite spermatheca (Fig 2G and 2G'). We also note that a functional P*adm-2*::ADM-2::GFP multicopy reporter (Fig 1F) displayed strongest expression in neurons where it accumulated at or near the plasma membrane (S3G Fig). Although these findings are consistent with ADM-2 acting in the major epidermis to affect molting, it is possible that ADM-2 could also affect molting through a nonautonomous mechanism. In addition, our findings suggest that ADM-2 likely functions in other tissues to affect processes other than molting.

## ADM-2 is trafficked through endo-lysosomal compartments and is sensitive to NEKL activities

To determine the identity of ADM-2 puncta, vesicles, and tubular-like structures in the epidermis, we performed colocalization experiments first using a CRISPR-tagged clathrin heavy

chain marker, GFP::CHC-1 [11]. Although statistically insignificant, occasional colocalization between ADM-2::mScarlet and apical clathrin was detected (Figs 3A–3C' and S4A). We note that endogenous plasma membrane–localized ADM-2::GFP and ADM-2::mScarlet both presented with extremely faint signals within hyp7 (Fig 2A and 2C, respectively), making detection and colocalization of this population difficult to assay relative to other compartments; this suggests that ADM-2 may be rapidly turned over at the plasma membrane, either by CME or through a CME-independent mechanism. We note that mammalian ADAMs, including the meltrin family, are internalized via CME [57–60]. In addition, we observed little or no colocalization between ADM-2::mScarlet and medial GFP::CHC-1 clathrin-containing structures (S4C–S4E' Fig), which may represent clathrin-coated vesicles emanating from the trans Golgi or endosomes.

In contrast, ADM-2::mScarlet exhibited partial colocalization with the endosomal marker $P_{hyp7}$::*hgrs-1*::*GFP* in both the sub-apical and medial planes (Figs 3D–3F', S4A, S4B, S4F–S4H and S4F'–S4H'). HGRS-1/HRS localizes to early endosomes and multivesicular bodies and is a component of the ESCRT-0 complex, which, together with ESCRT-I–III, promotes cargo sorting and lysosomal targeting [61–64]. Consistent with this, medial ADM-2::mScarlet showed strong co-localization within the lysosomal marker LysoTracker Green during intermolts (Fig 3G–3I), when lysosomes appear roughly spherical, and during molting periods (Fig 3J–3L), when lysosomes acquire a tubular morphology [65]. Thus, following rapid uptake into endosomes, ADM-2 is likely degraded by lysosomes, although some portion may be recycled back to the plasma membrane. Degradation of ADM-2 by lysosomes is further supported by the relative absence of medial ADM-2::GFP accumulation (Fig 2D), as GFP is acid sensitive and fluorescence is rapidly quenched within maturing endosomes and lysosomes [65,66].

Given our previous observations showing that NEKL–MLT proteins are required for normal trafficking within hyp7 [8,10,11], we tested if depletion of NEKLs caused changes in the abundance and subcellular localization of ADM-2. Notably, we observed total levels of ADM-2::mScarlet to increase slightly in auxin-treated *nekl-2*::*AID* adults, with more robust changes occurring in auxin-treated *nekl-3*::*AID* animals (Fig 3M–3P). Consistent with this, partial knockdown of MLT-3 in adults by RNAi led to modest increases in the levels of ADM-2::mScarlet and GFP::CHC-1 (S5A and S5B Fig). In worms that had undergone NEKL::AID depletion, ADM-2::mScarlet increased slightly in total levels and accumulated in large internal endocytic or lysosome-like structures (Fig 3M–3P). Collectively, these findings indicate that NEKL activities impact ADM-2 trafficking within hyp7.

## Suppression by ADM-2 occurs via a mechanism that is distinct from previous *nekl* suppressors

Loss of AP2 clathrin-adapter subunits and loss of the AP2 activator FCHO-1 individually suppress strong and/or null mutations in NEKLs and MLTs through their effects on CME [11]. We therefore tested if an *adm-2* null mutation could also suppress strong loss-of-function alleles of *nekls* and *mlts*. Notably, loss of *adm-2* was unable to restore viability to *nekl-2(fd91)*, *nekl-3(sv3)*, or *mlt-4(sv9)* strong loss-of-function alleles, which typically arrest as partially constricted L2/L3 larvae (Fig 1G). In contrast, knockdown of the sigma subunit of the AP2 complex led to ~60–85% viability in these genetic backgrounds [11]. These results suggest that the mechanisms underlying the suppression of molting defects by *adm-2* and CME-associated trafficking factors may be distinct.

To directly test if loss of ADM-2 can suppress CME defects in *nekls*, we examined the localization of GFP-tagged LRP-1/megalin, an apical transmembrane cargo that is trafficked via CME and is required for molting [2,5–7]. Using the auxin-inducible degron (AID) system to

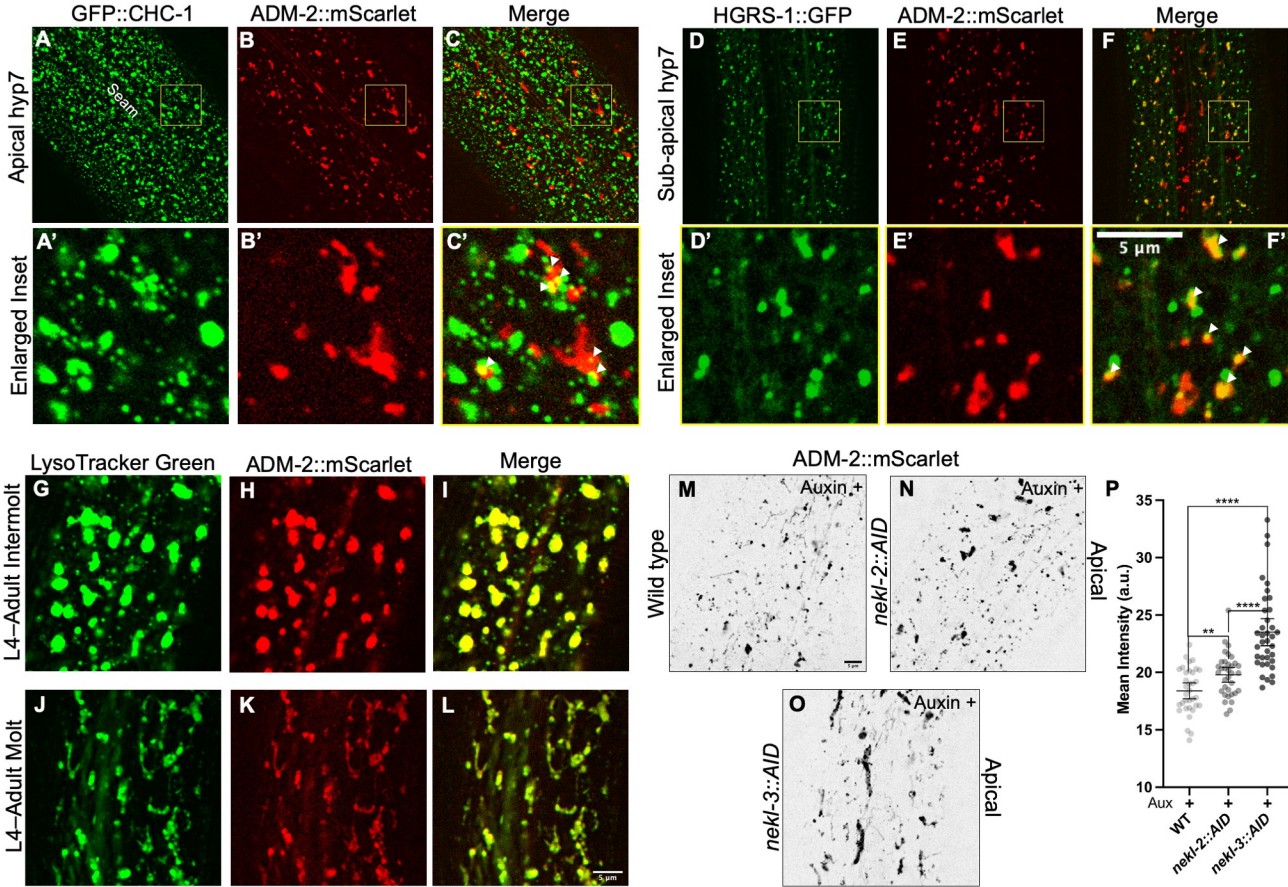

**Fig 3. ADM-2 localization to endocytic compartments is affected by NEKLs.** (A–L) Co-localization analysis of ADM-2::mScarlet with GFP::CHC-1 (A–C), P$_{hyp7}$::HGRS-1::GFP (D–F), and LysoTracker Green (G–L) within apical (A–C) and sub-apical or medial (D–L) planes of hyp7. A'–F' are insets of A–F confocal images. White arrowheads in C' and F' indicate colocalized puncta. For LysoTracker studies (G–L), representative confocal images during intermolt (G–I) and molting (J–L) stages are shown. (M–P) Apical hyp7 ADM-2::mScarlet localization in auxin-treated wild-type (M), *nekl-2::AID* (N), and *nekl-3::AID* (O) 2-day-old adults with average mean intensity calculations (P). Group means along with 95% confidence intervals (error bars) are indicated. p-Values were obtained by comparing means using an unpaired t-test: ****p $\leq$ 0.0001, **p $\leq$ 0.01. Bar in M = 5 µm (for A–F and M–O); in F' = 5 µm (for insets A'–F'); in L = 5 µm (for G–L). Raw data for this figure is provided in S1 File.

remove NEKL-3 activity in 1-day-old adult worms [11], we observed a dramatic accumulation of LRP-1::GFP at or near the apical membrane (Fig 4A and 4D), consistent with our previous findings [11]. As anticipated, loss of FCHO-1 partially corrected LRP-1::GFP mislocalization defects in NEKL-3::AID-depleted worms (Fig 4B and 4D), consistent with the ability of *fcho-1* mutations to suppress *nekl*-associated clathrin localization and mobility defects [11]. In contrast, loss of *adm-2* failed to correct LRP-1::GFP defects in NEKL-3::AID-depleted adults and surprisingly showed slightly enhanced apical LRP-1::GFP accumulation relative to *adm-2*+ NEKL-3::AID-depleted worms (Fig 4C and 4D; also see below). Collectively, these results indicate that loss of *adm-2* function does not suppress *nekl* molting defects by correcting CME deficits and is therefore likely to act through a distinct mechanism.

We recently reported that molting defects–but not trafficking defects–in *nekl-2; nekl-3* strains could be suppressed by induction of the *C. elegans* dauer pathway [29]. Similar to *adm-2*, suppression by the dauer pathway is effective in *nekl-2; nekl-3* double mutants but not in stronger loss-of-function *nekl* alleles. We therefore tested if *nekl-2; nekl-3* suppression by *adm-2* could be reversed by a reduction in dauer-pathway function. Our findings, however, indicate

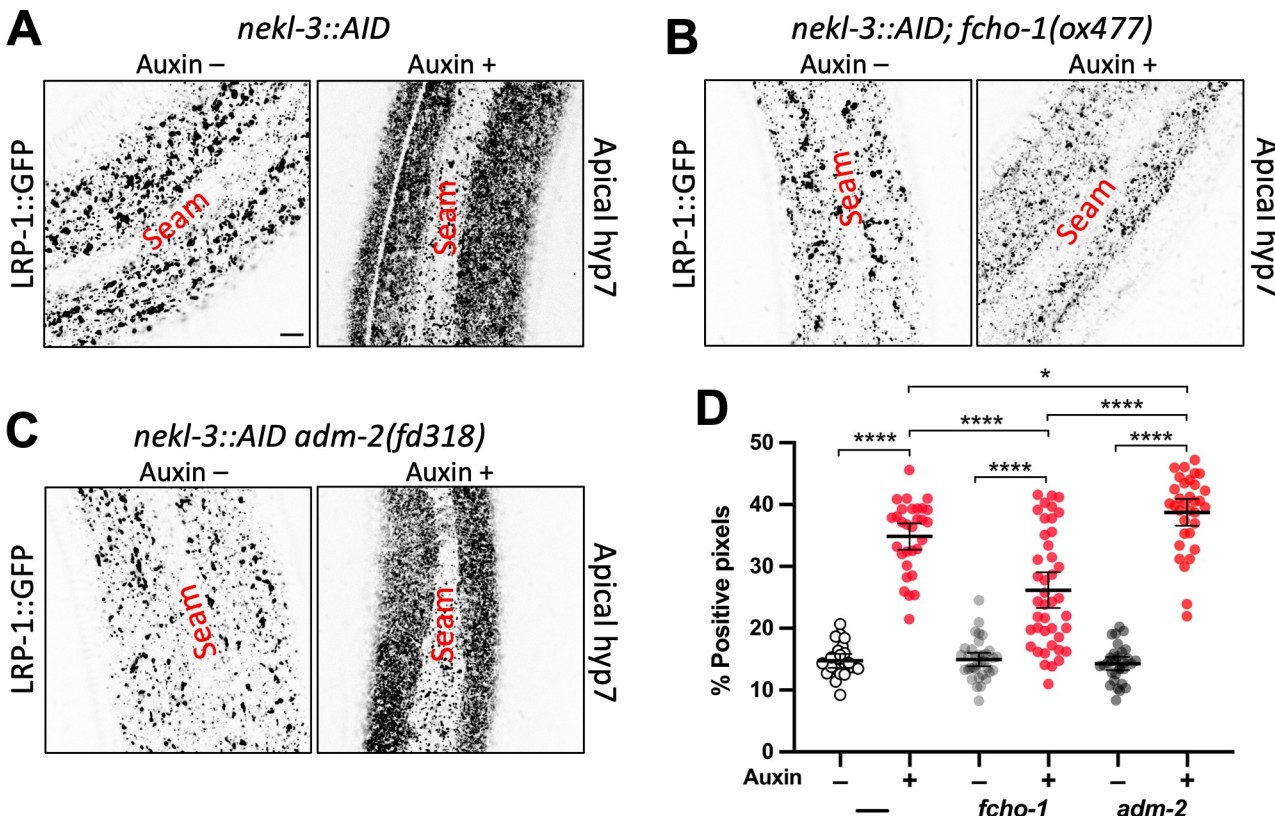

**Fig 4. Loss of *adm-2* function does not correct the *nekl* trafficking defects.** (A–C) Representative confocal images of 2-day-old adult worms expressing LRP-1::GFP in the apical hyp7 region of the epidermis. LRP-1 expression in the *nekl-3::AID* (A), *nekl-3::AID; fcho-1(ox477)* (B), and *nekl-3::AID; adm-2(fd318)* (C) mutant backgrounds in the absence (–) and presence (+) of auxin treatment. Bar in A = 5 μm (for A–C). (D) Dot plot showing the percentage of GFP-positive pixels within the apical plane of the worm epidermis for individuals of the specified genotypes and auxin treatment groups. Group means along with 95% confidence intervals (error bars) are indicated. p-Values were obtained by comparing means using an unpaired t-test: ****p ≤ 0.0001, *p ≤ 0.05. Raw data for this figure is provided in S1 File.

that *adm-2* suppression is not dependent on the dauer pathway, indicating that *adm-2* suppresses *nekl-2; nekl-3* mutants by a novel mechanism (S6 Fig).

## ADM-2 is a negative regulator of LRP-1

Our observation that LRP-1::GFP was further upregulated in NEKL-3::AID-depleted worms containing mutant *adm-2* (Fig 4D) suggested that ADM-2 may negatively regulate LRP-1. Notably, two mammalian homologs of LRP-1 (LRP1/LRP2) are regulated by matrix metalloproteinases and ADAM family members, including the meltrin ADAM12 [67–77]. We therefore hypothesized that ADM-2 may function as a sheddase for LRP-1. To test this possibility, we examined levels of LRP-1::GFP and LRP-1::mScarlet in *adm-2* null mutants and observed apical levels of LRP-1::GFP and LRP-1::mScarlet to be ~1.6-fold higher in *adm-2* mutants relative to wild type (Fig 5A–5F). We also observed a modest but statistically significant upregulation of LRP-1::GFP in *adm-2* mutant L4 larvae (S7A Fig). In contrast, loss of *adm-2* did not alter the levels of a clathrin heavy chain reporter at the apical membrane (S8A–S8D Fig), indicating that ADM-2 does not generally affect CME.

To further examine the relationship between ADM-2 and LRP-1 we tested the effects of heat-shock induced ADM-2 overexpression on LRP-1::GFP levels. One-day-old adult worms carrying the $P_{hsp-16}$::*adm-2* transgene and LRP-1::GFP were heat shocked and then allowed to

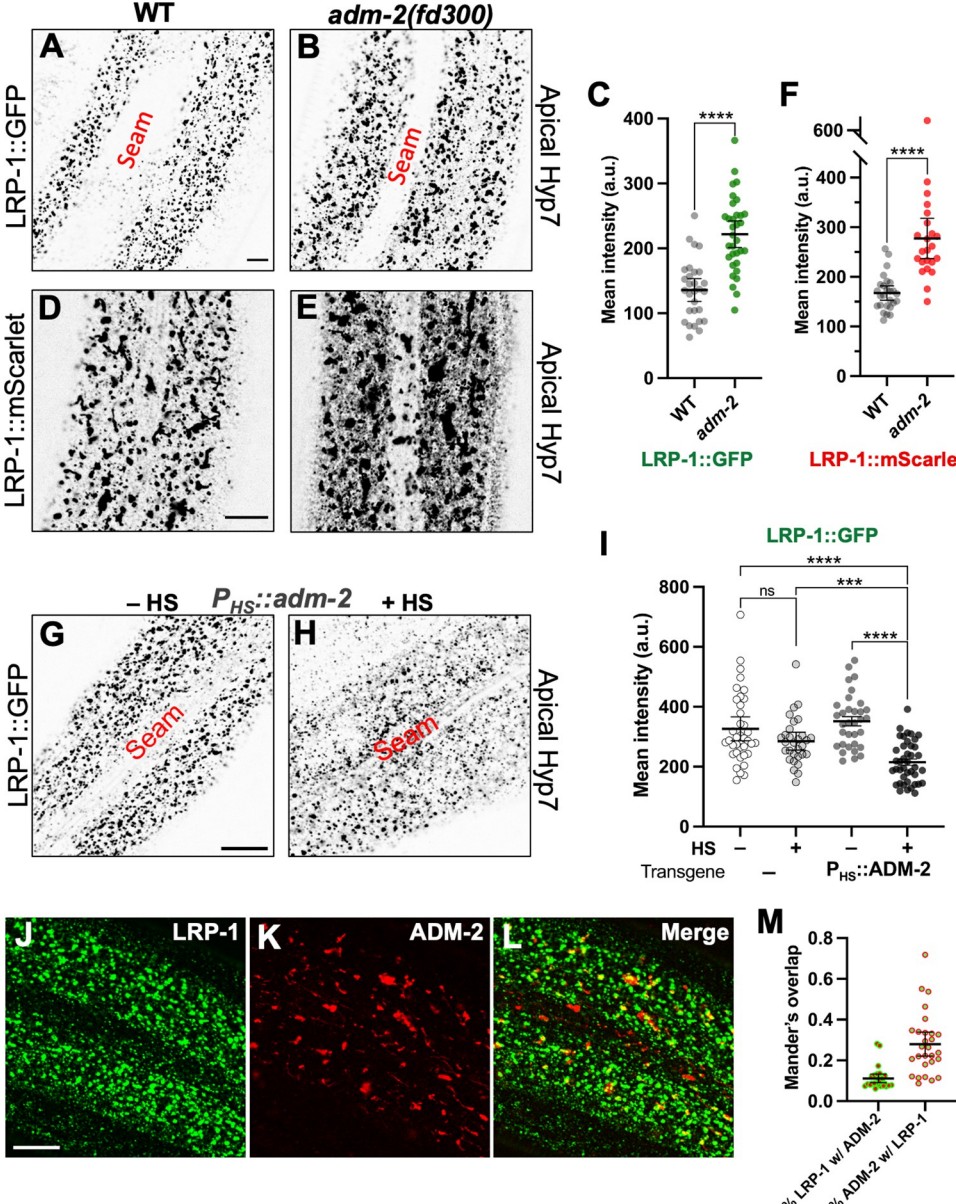

**Fig 5. ADM-2 activity affects LRP-1 levels.** Representative confocal images of 1-day-old adult wild-type (A, D) and *adm-2(fd300)* (B, E) worms expressing LRP-1::GFP (A, B) and LRP-1::mScarlet (D, E) in the hyp7 region of the hypodermis. Bar in A = 5 μm (for A, B). Bar in D = 10 μm (for D, E). Dot plot showing LRP-1::GFP (C) and LRP-1:: mScarlet (F) mean intensity (a.u.) within the apical plane of the worm hypodermis for each individual worm of the specified genotype. Representative confocal images of 1-day-old adult worms expressing LRP-1::GFP in the hyp7 region of the hypodermis. (G, H) LRP-1 expression in worms containing the $P_{hsp-16}$::*adm-2* transgene. Worms are shown in the absence of heat shock (G) and after heat shock (H). Bar in I = 10 μm (for I–L). (I) Dot plot showing LRP-1::GFP mean intensity (a.u.) within the apical plane of the worm hypodermis for each individual worm of the specified genotype and heat shock conditions. In C, F and I, group means along with 95% confidence intervals (error bars) are indicated. p-Values were obtained by comparing means using an unpaired t-test: ****p ≤ 0.0001; ***p ≤ 0.001; ns, p ≥ 0.05. (J–L) Co-localization analysis of LRP-1::GFP with ADM-2::mScarlet within apical plane of hyp7. (M) Dot plot showing quantification of Mander's overlap coefficient for the overlap of LRP-1::GFP with ADM-2::mScarlet proteins within the apical plane. Means along with 95% confidence intervals (error bars) are indicated. Raw data for this figure is provided in S1 File.

recover for ~2 hr prior to assaying LRP-1::GFP. Whereas heat shock did not substantially alter LRP-1::GFP levels in the absence of the $P_{HS}$::*adm-2* transgene, levels of LRP-1::GFP were ~1.6-fold lower when worms containing the $P_{HS}$::*adm-2* transgene were heat shocked relative to non-heat-shocked controls (Fig 5G–5I). Upregulation of ADM-2 in our system was supported by our observation that heat shock strongly induced GFP expression in a $P_{hsp-16}$::*adm-2*::*GFP* control strain (S7D–S7E Fig). Overall, our reciprocal findings are consistent with ADM-2 functioning as a negative regulator of LRP-1 and suggest that suppression of molting defects by loss of ADM-2 may occur in part through the upregulation or de-repression of cargo including LRP-1. Consistent with this possibility, we detected partial co-localization of LRP-1::GFP and ADM-2::mScarlet within hyp7 (Fig 5J–5M).

## Negative regulation of LRP-1 by ADM-2 does not require ADM-2 proteolytic activity

To further test the possibility that ADM-2 may function as a sheddase for LRP-1 we examined endogenously tagged LRP-1::mScarlet by western blot using an anti-RFP antibody in wild type and *adm-2* mutants. Interestingly, we observed that ~90% of LRP-1::mScarlet is apparently processed by proteolytic cleavage in wild-type worms (Fig 6A). Namely, the majority of LRP-1::mScarlet is present as an ~71 kD peptide whereas only a small proportion (~10%) is represented by the expected full-length product (~550 kD). We believe it unlikely that this result is due to an artifact of our methodology as we never detected additional (proteolytic) breakdown products of LRP-1::mScarlet and the efficient transfer of peptide markers ranging from ~30–460 kD was consistently observed. These observations suggest that LRP-1 is cleaved by a peptidase at approximate amino-acid position 4,350 of the 4,753 aa peptide, leaving only several hundred N-terminal amino acids of the ectodomain, the TM domain, and the short (157 aa) C-terminal domain.

Somewhat unexpectedly, we observed an identical banding pattern of LRP-1::mScarlet in *adm-2* null mutants (Fig 6A). Namely, the percentage of presumptive full-length LRP-1::mScarlet in wild type was 9.9% (95% CI, 5.0–14.9%; n = 8) versus ~11.6% (95% CI, 5.7–17.4%; n = 8) in *adm-2* mutants, suggesting that ADM-2 does not regulate LRP-1 through proteolysis. To further test this possibility we compared LRP-1::GFP levels in wild-type and CRISPR-generated *adm-2* Zn-MTP mutant worms (Fig 6B). Notably, loss of ADM-2 metalloproteinase function did not alter total levels of LRP-1::GFP in the apical region of hyp7, consistent with ADM-2 regulating LRP-1 through a non-proteolytic mechanism. As an additional test, we examined LRP-1::GFP levels in worms following heat-shock induction of a metalloproteinase-defective variant of ADM-2 ($P_{HS}$::ADM-2–MTP; Fig 6C). Notably, we found that overexpression of ADM-2–MTP decreased levels of LRP-1::GFP to a similar extent as wild-type ADM-2 ($P_{HS}$::ADM-2), in contrast to worms containing either no array (−) or an array containing an empty-vector heat-shock plasmid ($P_{HS}$::EV) (Figs 5I; 6C, 6D and S7B, S7C). We also note that although western blotting suggested a ~1.7-fold increase in total levels of LRP-1::mScarlet in *adm-2* mutants relative to wild type, this difference was only of marginal statistical significance (Wilcoxon Signed Rank Test, p = 0.096; n = 8). Taken together, our findings indicate that ADM-2 can act as a negative regulator of LRP-1, although it appears to do so through a mechanism that is independent of its metalloproteinase function.

## Multiple domains likely contribute to the effect of ADM-2 on molting

Our above results suggest that while the metalloproteinase domain of ADM-2 plays a role in its regulation of the molting process, other domains may also make important contributions. To further examine this we tested CRISPR variants designed to disrupt additional domains of

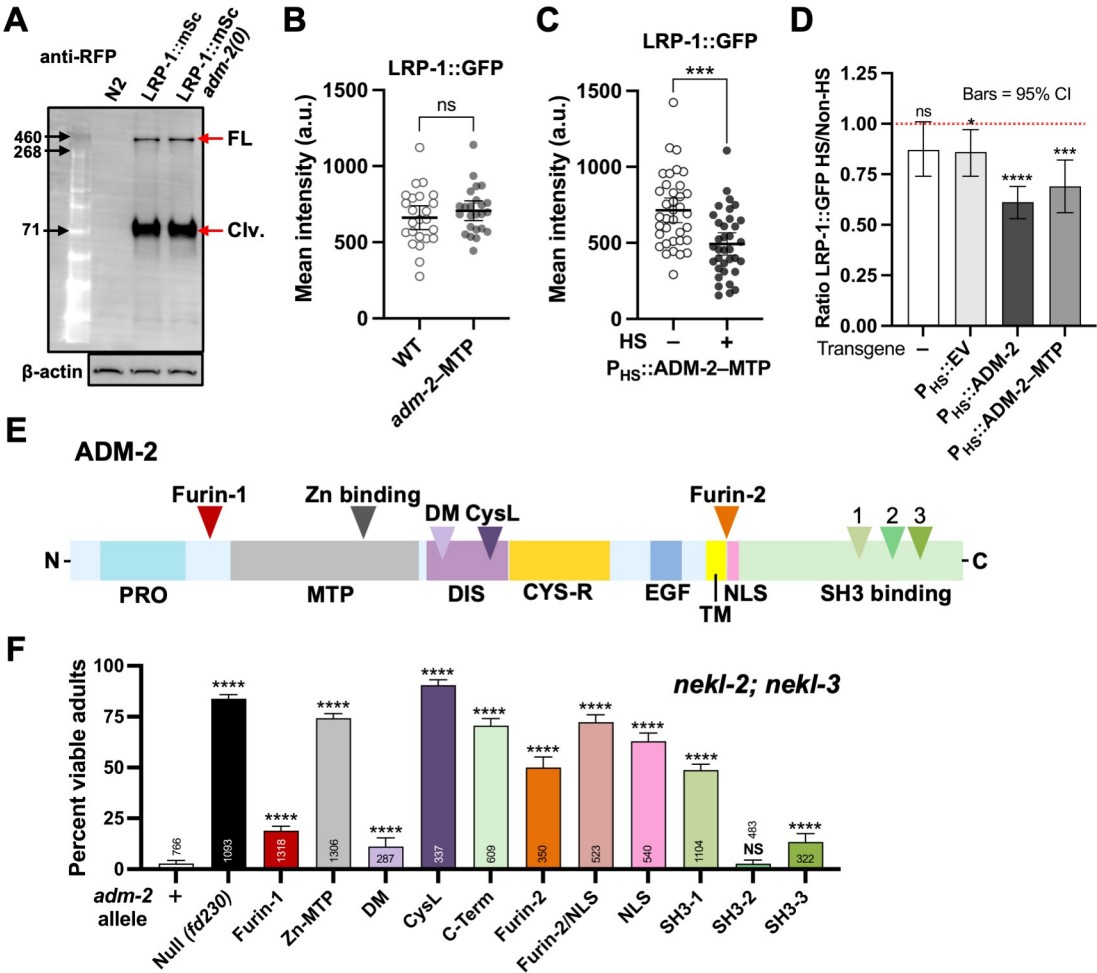

**Fig 6. Regulation of molting by multiple ADM-2 domains.** (A) Western blot showing RFP antibody recognizing the mScarlet tag in the endogenously expressed LRP-1::mScarlet protein. This blot shows the (From left) high molecular weight ladder, total protein lysates from Wild Type (N2), LRP-1::mScarlet and LRP-1::mScarlet; *adm-2(fd300)* strains. Full length (FL) protein and the cleaved cytoplasmic peptide (Clv.) shown in red arrows. Corresponding control blot recognizing β-actin is shown below. (B, C) Dot plots showing the LRP-1::GFP mean intensity (a.u.) within the apical plane of the worm hypodermis for each individual worm in; (B) Wild Type and *adm-2* metalloprotease mutant background; (C) worms containing the $P_{hsp-16}$::*adm-2 (metalloprotease mutant)* transgene in the absence of heat shock and after heat shock. In B and C, group means along with 95% confidence intervals (error bars) are indicated. p-Values were obtained by comparing means using an unpaired t-test: ***p ≤ 0.001; ns, p ≥ 0.05. (D) Bar plot showing the ratio of LRP-1::GFP mean intensity in the absence of heat shock and after heat shock in the worms containing no transgene, $P_{hsp-16}$::*Empty Vector*, $P_{hsp-16}$::*adm-2* and $P_{hsp-16}$::*adm-2(metalloprotease mutant)* transgenes respectively. Error bars represent 95% confidence intervals. p-Values were obtained by comparing means in the absence of heat shock and after heat shock using an unpaired t-test: ****p ≤ 0.0001; ***p ≤ 0.001; ns, p ≥ 0.05. (E) Schematic representation of predicted protein domains within ADM-2. PRO, prodomain; MTP, metalloprotease domain; DIS, disintegrin domain; CYS-R, cysteine repeat region; TM, transmembrane domain; NLS, predicted nuclear localization signal; DM, disintegrin motif; CysL, cysteine loop; SH3 binding (1–3) Src homology 3 binding domains. (F) Percentage of viable (non-molting-defective) *nekl-2; nekl-3* adults in the indicated CRISPR-derived *adm-2* mutant backgrounds. Error bars represent 95% confidence intervals. p-Values were determined using Fisher's exact test: ****p ≤ 0.0001; ns, not significant. Raw data for this figure is provided in S1 File.

ADM-2. Among the N-terminal variants, we observed strong suppression of *nekl-2; nekl-3* molting defects by mutations predicted to disrupt a cysteine loop within the disintegrin domain (CysL; Figs 6E–6F and S9A–S9C). In addition, weak but above-background suppression was observed with mutations affecting a predicted Furin-cleavage site (Furin-1) and within a conserved disintegrin motif (DM; Figs 6E–6F and S9A–S9C). We also tested the

importance of the ADM-2 C-terminal domain (R696–K952) using CRISPR-generated lines. An in-frame deletion of most of the C terminus (CT) of ADM-2 led to strong suppression of *nekl-2; nekl-3* molting defects, whereas perturbation of individual SH3-binding domains led to suppressive effects ranging from strong (SH3-1), to moderate (SH3-3), to absent (SH3-2; Figs 6E–6F and S9A–S9C). Lastly, several tested variants affecting a region just C-terminal to the transmembrane (TM) also led to the suppression of *nekl-2; nekl-3* molting defects (Figs 6E–6F and S9AC). This region contains a predicted nuclear localization signal (NLS) that overlaps with a Furin-cleavage motif (Furin-2) and could conceivably be involved in the processing and nuclear translocation of ADM-2. Although the nuclear translocation of ADAM cytosolic domains has been reported [34, 78], we failed to detect nuclear ADM-2 in our expression studies.

While these mutational studies are consistent with the model that non-proteinase functions of ADM-2 may contribute to the suppression of *nekl-2; nekl-3* molting defects, several caveats should be noted. Most notable is that the domain-targeted variants may unintentionally influence the structure and function of other domains. In addition, mutations that affect the subcellular localization or stability of ADM-2 may also suppress molting defects (also see Discussion). Nevertheless, our cumulative findings suggest that functions independent of metalloproteinase activity may contribute to the suppression of *nekl* defects by *adm-2* loss of function.

## Discussion

In this study we identified a functional role for the sole *C. elegans* meltrin family member, ADM-2, in the molting process. Whereas the absence ADM-2 did not overtly impact the molting cycle, its loss of function, identified through a non-biased genetic screen, strongly suppressed molting defects in *nekl-2; nekl-3* mutants. Previous studies have implicated proteases as positive regulators of molting including NAS-36, NAS-37, CPZ-1, and SURO-1 [79–84]. Extracellular proteases have been suggested to be important for the detachment and degradation of the old cuticle and may also play a dynamic role in ECM remodeling during new cuticle synthesis [2]. In contrast, our data implicate ADM-2 as a negative regulator of the molting process. Interestingly, roles for protease inhibitors in molting have also been reported, as loss of the Kunitz domain–containing protease inhibitors MLT-11 and BLI-5 lead to molting defects [80,85,86]. Protease inhibitors have been suggested to be important for temporally or spatially restricting the activity of extracellular proteases during the molting cycle. Consistent with this, epidermally expressed proteases and protease inhibitors are highly enriched among genes that are transcriptionally regulated with the molting cycle, indicating a tight control of proteolytic activity [87,88].

Although we previously reported the suppression of *nekl* molting defects by mutations affecting genes closely connected to CME, our data do not support a role for ADM-2 in the regulation of intracellular trafficking per se. Rather, our findings are consistent with ADM-2 being a cargo of CME, including during its passage through endosomes and its turnover in lysosomes. Moreover, we observed increased levels of ADM-2 after NEKL::AID knockdown along with the retention of ADM-2 in intracellular compartments of the epidermis. Collectively these findings suggest that loss of NEKL functions may lead to increased or aberrant ADM-2 activity, which may contribute to molting defects in these mutants.

Notably, we identified a positive regulator of molting, LRP-1, as a target of ADM-2 regulation. Epidermal LRP-1 levels were increased in *adm-2* null mutants and reduced when ADM-2 was overexpressed, indicating that ADM-2 negatively regulates LRP-1. In addition, a subset of LRP-1 endosomal puncta colocalize with ADM-2, suggesting that this regulatory interaction may occur within the trafficking pathway. Increased LRP-1 in *adm-2* mutants could contribute

to the suppression of *nekl-2; nekl-3* molting defects by increasing the uptake of steroid hormone precursors, a process that we have shown to be defective in *nekl* mutants [11,29]. Nevertheless, our data also indicate that ADM-2 is unlikely to regulate LRP-1 through its sheddase activity, as was previously demonstrated for human ADAM and LRP-1 homologs [67–77]. Rather, our findings suggest a non-canonical mechanism for the regulation of a low-density lipoprotein-like receptor by an ADAM family member. How this regulation may occur, along with understanding how specific domains of ADM-2 may contribute to molting control and other developmental processes, remains to be determined.

Consistent with the above, our CRISPR-based structure-function analysis of ADM-2 suggests that several different domains and functions may contribute to the suppression of *nekl-2; nekl-3* by *adm-2*. As mentioned previously, however, these findings could be due in part to "off-target" effects of our variants on other domains within ADM-2. As one example, modeling of the CysL variant using the AlphaFold database [89,90] and the Robetta protein structure prediction service [91,92] suggests that in addition to impacting the disintegrin domain, this variant could also affect the coordination of Zn binding by histidine residues within the MTP domain (S9D Fig). Likewise, it is possible that suppressing mutations within the MTP domain impact other N-terminal activities. In contrast, other variants, including the C-terminal SH3-1, are not predicted to affect MTP domain structure (S9E Fig) but could potentially affect the subcellular localization, stability, or other activities of ADM-2. In fact, a role for the C-terminus in mediating ADAM function would not be unexpected given that this domain is proposed to be important for the regulation of cell signaling and may contain motifs involved in the 'inside-out' regulation of ADAM metalloprotease activities [31,32,34,93].

Our cumulative findings, together with published data from other groups, point to a working model in which epidermal intracellular trafficking may play two roles in the molting process. On the one hand, endocytosis may be required for the internalization of factors that promote molting, such as recycled cuticle components [65,94] and cholesterol via LRP-1 [5,6], a function consistent with our observation that loss of *nekls* leads to defects in the transcription of hormonally regulated molting genes [29]. In addition, endocytosis may also promote the uptake and degradation of cargo that might otherwise exert an inhibitory effect on molting, a function that we propose for ADM-2. In fact, cargo in both categories may have functional interactions, such as the observed negative regulation of LRP-1 by ADM-2. In this model, loss of ADM-2 would lead to the de-repression of LRP-1, along with other potential effectors of molting, thereby compensating for a partial loss of NEKL trafficking functions. When NEKL functions are more severely reduced, however, loss of ADM-2 would be unable to offset the resulting deficiencies, such as strong defects in molting gene expression, consistent with our observation that *adm-2* mutations do not suppress strong loss-of-function alleles in *nekls*. In summary, our findings expand the roles for NEKLs and intracellular trafficking in the molting process and implicate an ADAM family member as a negative regulator of the molting process.

## Materials and methods

### Strains

*C. elegans* strains were maintained according to standard protocols [95] and were propagated at 22˚C, unless stated otherwise. Strains used in this study are listed in S1 Table.

### Transgenic rescue

Fosmids containing rescuing sequences for *adm-2*/C04A11.4 (WRM0620dD12, WRM0632aG02, and WRM0610cA04, 2–6 ng/µl each + *sur-5::RFP* [pTG96], 50–100 ng/µl)

were injected into strain WY1342. Stable strains (WY1386 and WY1388) containing rescuing arrays for both *nekl-3* (*fdEx286*; GFP⁺) and *adm-2* (*fdEx315* or *fdEx356*; RFP⁺) were scored to determine the percentage of viable RFP⁺ progeny.

## Determination of dominant versus recessive alleles

To distinguish between dominant and recessive alleles, we first crossed suppressed *nekl-2* (*fd81*); *nekl-3*(*gk894345*) *fd130* hermaphrodites to WY1145 [*nekl-2*(*fd81*); *nekl-3*(*gk894345*); *fdEx286* (*nekl-3*⁺ + *sur-5*::*GFP*)] males and scored for suppression of GFP⁻ cross-progeny males. For *fd130* and *fd162*, 50/96 and 30/52 viable cross-progeny adult males were GFP⁻, respectively, indicating that these mutations are either dominant or on LG X. We next crossed *nekl-2*(*fd81*); *nekl-3*(*gk894345*) *fd130* hermaphrodites to WY1232 [*nekl-2*(*fd81*); *nekl-3* (*gk894345*); *fdEx186* (*nekl-3*⁺ + *sur-5*::*GFP*); *fdEx197* (*sur-5*::*RFP*)] males and scored for suppression of GFP⁻RFP⁺ cross-progeny hermaphrodites. In the case of cross-progeny *fd130*/+ adult hermaphrodites, 62/62 were either GFP⁺ RFP⁻ or GFP⁺ RFP⁺; no GFP⁻RFP⁺ adults were observed. Similarly, 182/185 *fd162*/+ adult hermaphrodites were either GFP⁺ RFP⁻ or GFP⁺ RFP⁺, and only 3/185 were GFP⁻RFP⁺. Given the ~98.5% penetrance of *nekl-2*(*fd81*); *nekl-3* (*gk894345*) larval lethality [30], our results indicate that *fd130* and fd162 are fully recessive but are on LG X.

## RNAi

Primers containing the binding motif for T7 RNA polymerase (5'-TAATACGACTCAC TATAGGGAGA-3') and corresponding to *adm-2* (5'-GACCACAACAATGATACGGTC GAA-3'; 5'-CCTGGACACAATGCAGCATTTTGA-3'), *unc-71* (5'-TGTCGTCGACGGTTCC GAAGA-3'; 5'-GCATCAGACAGACCAGGCATAG-3'), *adm-4* (5'-ATGCATTCAATACAC GTGTGA-3'; 5'-CTTCCTCTCCCAGATATATCGT-3'), *sup-17* (5'-AGTGTCAACCTGGTC TTCCTG-3'; 5'-CTGTGCCCATTGTGTTAGAGTTTC-3'), *mig-17* (5'-CTCAGCTACAC AAGGAATGGC-3'; 5'-TTCGCACACGTTCTACAACA-3'), *tag-275* (5'-TGTTCTCGCGTC ATTCGTTGC-3'; 5'-ACTCGGTTTATTGGAACATTTGGC-3'), *F27D9.7* (5'-CAACATTC TGTGCGATGCGGT-3'; 5'-TTAAATGGGCGCGACAGATCC-3'), *adt-1* (5'-GTCAGTGC ACTCACTGGACAT-3'; 5'-GGTTAGGCATGGCCTGAATCT-3'), *adt-2* (5'-GAAGAC GAAACCGAAGTCTGC-3'; 5'-TTACCTCCCCATGCAGCATTT-3'), *adt-3* (5'-CAGGTA TGTAACGGTGACTCCA-3'; 5'-CATTACACATGGTCCGGTTTC-3'), and *gon-1* (5'- TGGA TCACTGAAGATGTGTCT-3'; 5'- GCACTCCAATCAGTATTTCTC-3') were used to generate dsRNA using standard methods [96]. After injection at 0.8–1.0 µg/µl into WY1145 hermaphrodites, F1 progeny were scored for adult viability. For RNAi feeding experiments, the relevant bacterial strains were obtained from Geneservice and IPTG (8 mM) was added to growth plates [97]. Worm strains were grown on *lin-35(RNAi)* plates for two generations to increase RNAi susceptibility [98]. Second-generation fourth larval stage (L4) worms growing on *lin-35(RNAi)* plates were transferred to experimental plates and were imaged after 48 hours. RNAi feeding experiments were performed at 20°C.

## ADM-2 CRISPR mutant alleles

Design of repair sequences containing introduced restriction sites was facilitated using CRISPRcruncher [99]. For details on primers sequences see S2 File.

**Strong loss-of-function alleles.** Alleles *fd228–fd230* and *fd235–fd237* were generated using guide dual sequences SB1 and SB2, PCR amplification primers SB3 and SB4, and the sequencing primer SB6. In *fd228–fd230* and *fd235–fd237*, an ~3.2-kb region spanning *adm-2a* exons 3–9 is deleted. *fd228* is an indel predicted to encode sequences through T122 of ADM-

2a, followed by eight divergent amino acids and a stop codon. *fd229* is an indel predicted to encode sequences through S123 of ADM-2a, followed by a stop codon. *fd230* is an indel predicted to encode sequences through T122 of ADM-2a, followed by a stop codon. *fd235* is an indel predicted to encode sequences through T122 of ADM-2a, followed by 23 new amino acids and a stop codon. *fd236* is an indel predicted to encode sequences through F118 of ADM-2a, followed by six new amino acids and a stop codon. *fd237* is an indel predicted to encode sequences through T122 of ADM-2a, followed by a single new amino acid and a stop codon. Alleles *fd298–302* were generated using guide sequences SB46 and SB47, PCR amplification primers SB48–SB51, and the sequencing primer SB52. In *fd298–fd302*, *fd313*, *fd316*, and *fd317* an ~7.4-kb region spanning *adm-2a* exons 1–19 is deleted. *fd298* is an indel that encodes sequences through T2 of ADM-2a followed by four new amino acids and a stop codon. *fd299* is a deletion that encodes sequences through M1 of ADM-2A, followed by three new amino acids and a stop codon. *fd300* is an indel that encodes sequences through D3 of ADM-2a, followed by 27 new amino acids and a stop codon. *fd301* and *fd302* are identical indels that encode sequences through D3 of ADM-2a, followed by 12 new amino acids and a stop codon. *fd313* encodes sequences through T2 of ADM-2a, followed by 40 new amino acids and a stop codon. *fd316* deletes the normal start codon; an alternative ATG is predicted to encode 24 new amino acids followed by a stop codon. *fd317* encodes sequences through D3 of ADM-2a, followed by 16 new amino acids and a stop codon. *fd318* encodes sequences through D3 of ADM-2a, followed by 8 new amino acids and a stop codon.

**adm-2 metalloprotease mutation (Zn-binding domain).** Alleles *fd243–fd247*, *fd391* were generated using the guide sequence SB27, the repair template Rep2, PCR amplification primers SB28 and SB29, and sequencing primers SB31 and SB32. *fd243–fd247*, *fd391* change the predicted ADM-2a Zn-metalloprotease active site spanning H312–H322 (**HELGH**TFG**M**D**H**) to **DA**LA**Y**TF**R**MD**Y** (altered aa are in bold). The Zn-metalloprotease consensus motif is HEXXHXUGUXH, where U is an amino acid containing a bulky hydrophobic residue. The edited locus contains an introduced BstBI site.

**adm-2 disintegrin motif mutation.** Allele *fd322* was generated using guide sequence DF1, the repair template Rep9, PCR amplification primers DF2 and DF3, and sequencing primers DF4 and DF5. *fd322* changes the predicted ADM-2a disintegrin motif (DM) spanning E388–G396 (EPGE**ECDCG**) to EPGE**VLADP**. The edited locus contains introduced NheI and BamHI sites.

**adm-2 cysteine loop mutation.** Alleles *fd324–fd325* were generated using guide sequence DF6, the repair template Rep10, PCR amplification primers DF2 and DF3, and sequencing primers DF4 and DF5. *fd324–fd325* change the predicted ADM-2a cysteine loop (CysL) spanning C438–P459 (CRAAIGICDL**DEYCNG**ETNDCP) to CRAAIGICDL**QQNGDH**ETNDCP. The edited locus contains an introduced PstI site.

**adm-2 furin 1 mutation.** Alleles *fd288-fd290* were generated using guide sequence SB17, the repair template Rep6, PCR amplification primers SB18 and SB19, and sequencing primers SB20 and SB21. *fd288-fd290* change the predicted ADM-2a furin-1 cleavage site spanning R149–R152 (**R**KK**R**) to **V**KK**V**. The edited locus contains an introduced BamHI site.

**adm-2 SH3-binding domain 1 mutation.** Alleles *fd248–fd251* were generated using the guide sequence SB32, the repair template Rep3, PCR amplification primers SB16 and SB34, and sequencing primer SB35. *fd248–fd251* alter the predicted ADM-2a SH3-1 domain spanning V722–P731 (**VP**V**RKA**PPPP) to **EG**V**LAA**GAVG. The edited locus contains an introduced XhoI site.

**adm-2 SH3-binding domain 2 mutation.** Alleles *fd252–fd256* were generated using the guide SB36, the repair template Rep4, PCR amplification primers SB37 and SB38, and sequencing primers SB39 and SB40. *fd252–fd256* alter the predicted ADM-2a SH3-binding

domain 2 domain spanning P839–V853 (**P**NVQ**PPP**V**PRP**S**DDV**) to **G**NVQ**GAG**V-**GAG**SLLE. The edited locus contains an introduced XhoI site.

*adm-2* **SH3-binding domain 3 mutation.** Alleles *fd257–fd259* were generated using the guide sequence SB41, the repair template Rep5, PCR amplification primers SB37 and SB38, and sequencing primers SB39 and SB40. *fd257–fd259* alter the predicted ADM-2a SH3-binding domain 3 spanning K874–K884 (**K**TL**P**L**PPP**L**PK**) to **I**TL**E**L**GAG**L**GL**. The edited locus contains an introduced XhoI site.

*adm-2* **C-terminal (cytoplasmic domain) deletion.** Alleles *fd231–fd234* were generated using the guide sequences SB12 and SB13, the repair template Rep1, PCR amplification primers SB14 and SB15, and sequencing primer SB16. *fd231–fd234* remove sequences from H718 to M942 of ADM-2a and introduce a XhoI site.

*adm-2* **furin-2 mutation.** Alleles *fd292* and *fd293* were generated using the guide sequence SB22, the repair template (Rep7), PCR amplification primers SB23 and SB24, and sequencing primers SB25 and SB26. *fd292* and *fd293* alter the predicted ADM-2a N-terminal furin cleavage site spanning R696–R699 (**R**VK**R** to **V**VK**L**). The edited locus contains a new AflII site.

*adm-2* **furin-2/NLS mutation.** Alleles *fd310–fd312* were generated using the guide sequence SB22, the repair Rep8, PCR amplification primers SB53 and SB54, and sequencing primers SB55 and SB56. *fd310–fd312* alter the predicted ADM-2a monopartite nuclear localization signal (NLS) spanning Y694–V704 (YYRVKRKRNLV to **VVL**V**GAIA**NLV) as well as the adjacent predicted bipartite NLS spanning R696–D716 (RVKRKRNLVSEWWSVVKKKFD to **VVL**V**GAIA**NLV**Y**EWWSVVKKKFD). The edited locus contains a new AccI site.

*adm-2* **NLS mutation.** Allele *fd326* were generated using the guide sequence SB22, the repair Rep11, PCR amplification primers SB53 and SB54, and sequencing primers SB55 and SB56. *fd326* alter the predicted ADM-2a monopartite nuclear localization signal (NLS) spanning Y694–S705 (YYRVKR**KRNLVS** > YYRVKR**EDPGDP**). The edited locus contains a new XmaI site.

## Detailed sequence information on all *adm-2* alleles can be found in S1 File and S3 File

**ADM-2 expression plasmids and strains.** *adm-2*::*GFP* and *adm-2*::*mScarlet* endogenously tagged strains were made using CRISPR/Cas9 technology in collaboration with Suny-Biotech Corporation (China). Expression vectors were generated by amplifying the *adm-2* promoter region from fosmid WRM0620dD12 using primers SB57 and SB58. After digestion with SphI and SalI, the ~2.1-kb PCR product was inserted into pPD95.75 to create pDF403–pDF405. *adm-2* cDNA was amplified from plasmid pDONR201 (Horizon Inc.) using primers SB59 and SB60. Digestion and ligation of the ~2.8-kb PCR product and pDF403 with XmaI and KpnI generated pDF417–pDF420, which were confirmed by sequencing. pDF417 (~100 ng/μl) was injected into N2 worms with *sur-5*::*RFP* (~50 ng/μl) to obtain lines carrying extrachromosomal arrays (*fdEx353*, *fdEx354*). CRISPR-tagged *adm-2*::*mScarlet* and *adm-2*::*GFP* strains contain codon optimized fluorescent-reporter insertions just preceding the *adm-2a* stop codon (SUNY biotech). In addition, sequences upstream of the stop codon contain the indicated (bold) silent mutations (LG X 13695343–13695387).

TC**T**GAAGATGCAGCTGCAACCGAAGAAAAAGTAGATGTTCGCTC**C** (wild type)
TC**A**GAAGATGCAGCTGCAACCGAAGAAAAAGTAGATGTTCGCTC**G** (CRISPR tagged)

*adm-2* **and** *adm-2*::*GFP* **heat shock strains.** An ~4.6-kb *adm-2*::*GFP*::*unc-54* 3'UTR cDNA product was obtained by digesting pDF420 with XbaI and ApaI enzymes. Digestion of

heat shock vectors pPD49.78 and pPD49.83 was performed using NheI and ApaI to obtain an ~3-kb vector backbone. Ligation of the *adm-2* cDNA with vector backbones pPD49.83 and pPD49.78 generated pDF429–pDF431 and pDF432–pDF434, respectively. Likewise, an ~2.8-kb *adm-2*::*GFP* product was obtained by digesting pDF420 with XbaI and KpnI enzymes. Digestion of heat shock vectors pPD49.78 and pPD49.83 was performed using NheI and KpnI to obtain an ~3-kb vector backbone. Ligation of *adm-2*::*GFP* with vector backbones pPD49.83 and pPD49.78 generated pDF423–pDF425 and pDF426–pDF428, respectively. pDF430 and pDF433 (50 ng/μl each) were injected into N2 with pRF4 [*rol-6(gf)*] (~50 ng/μl) to obtain lines carrying extrachromosomal arrays (N2: *fdEx373–375*). Likewise, pDF424 and pDF425 (50 ng/μl each) were injected into N2 with pRF4 [*rol-6(gf)*] (~50 ng/μl) to obtain lines carrying extra-chromosomal arrays (*fdEx381*, *fdEx382*).

**Heat shock methods.** For Figs 5G–5I and 6C and S7B–S7D 1-day-old adult worms grown at 20˚C were heat shocked at 34˚C for 4 hours and were shifted to 20˚C for 2–3 hours before imaging.

**Protein domain identification and alignment tools.** The following sites were used to identify domains within ADM-2 and human ADAM homologs:

http://nls-mapper.iab.keio.ac.jp/cgi-bin/NLS_Mapper_form.cgi
https://www.ncbi.nlm.nih.gov/Structure/cdd/wrpsb.cgi
https://www.ebi.ac.uk/interpro/
https://prosite.expasy.org/
https://psort.hgc.jp/form2.html
http://www.cbs.dtu.dk/services/TMHMM/
http://www.cbs.dtu.dk/services/SignalP/
http://phobius.sbc.su.se/
http://tcoffee.crg.cat/apps/tcoffee/do:regular
https://embnet.vital-it.ch/software/BOX_form.html

**Image acquisition.** Fluorescence images in the following figure panels—Figs 4A–4C; 5A, 5B; S3G and S8—were acquired using an Olympus IX81 inverted microscope with a Yokogawa spinning-disc confocal head (CSU-X1). Excitation wavelengths were controlled using an acousto-optical tunable filter (ILE4; Spectral Applied Research). MetaMorph 7.7 software (MetaMorph Inc.) was used for image acquisition. z-Stack images were acquired using a 100×, 1.40 N.A. oil objective. The rest were acquired using an Olympus IX83 inverted microscope with a Yokogawa spinning-disc confocal head (CSU-W1). z-Stack images were acquired using a 100×, 1.35 N.A. silicone oil objective. cellSense3.1 software (Olympus corporation) was used for image acquisition. DIC images in the panels Figs 1A–1C and S7D were acquired using a Nikon Eclipse epiflourescence microscope and the cellSense3.1 software (Olympus corporation).

**Image analysis.** Mean intensity (measured in arbitary units, a.u.), percent of fluorescence-positive pixels above threshold, and the colocalization analysis were performed using Fiji software (NIH; available at https://imagej.net/Fiji/Downloads). For a given z-plane of interest, rolling ball background subtraction was performed (radius = 50 pixels), and the polygon selection tool was used to choose the region of hyp7 in which the mean intensity was quantified (Figs 5, 6, S5, S7 and S8). The percentage of fluorescence-positive pixels for the region of interest (Figs 4 and S8) was determined after thresholding, and the "Huang" thresholding algorithm was used for strain comparisons. For colocalization, rolling ball background subtraction was performed (radius = 25 pixels), followed by use of the mean filter (radius = 2 pixels) to minimize noise. Finally, the same thresholding algorithm was used for one particular channel to obtain binary images to be used as masks (GFP::CHC-1 and HGRS-1::GFP —"Otsu"; ADM-2:mScarlet—"Isodata"; LRP-1::GFP–"Li"). This binary mask was combined

using the "AND" boolean operation to the original image and the combined image was used for the colocalization analysis. Data from Fig 4A–4C (Olympus IX81) and Figs 5G–5I, 6B, 6C and S7 (Olympus IX83) were obtained on different confocal microscopes and thus differ in their mean intensity values.

**Auxin treatment.** Auxin (indole-3-acetic acid) was purchased from Alfa Aesar. A 100× stock auxin solution (0.4 M) was made by dissolving 0.7 g of auxin in 10 ml of 100% ethanol. A mixture of 25 μl of stock auxin solution and 225 μl of distilled water was added to plates containing 1-day-old adult worms.

**Protein 3D structure analysis.** PDB file (Identifier: AF-G5EDW5-F) containing the three-dimensional structure details of *C. elegans* ADM-2 was obtained from the AlphaFold database (https://alphafold.ebi.ac.uk/) [89, 90]. Using the AlphaFold structure as template, homology modeling was performed by the online Robetta structure prediction server (https://robetta.bakerlab.org/) to obtain the predicted the three-dimensional structures of the respective ADM-2 mutants [91, 92]. For modeling CM (Comparative modeling) option was used and the number of models to sample was selected as 1. Other options remained unchanged. The homology modeled three-dimensional structures were rendered, and was superimposed onto the AlphaFold structure of ADM-2 using the PyMOL 2 software (The PyMOL Molecular Graphics System, Version 2.0 Schrödinger, LLC.).

**Western blot analysis.** 200 L4 worms of each strain was picked into 50 μl of urea lysis buffer (7M urea, 1M thiourea, 2% CHAPS, 30mM Tris pH 8.0). 0.5 μl of protease inhibitor cocktail (Thermo Scientific: 1861280) was added and homogenized using a hand-held homogenizer. After homogenization, samples were spun at 8000 rpm (4°C) for 5 mins. Supernatant was carefully transferred into a fresh tube. 40 μl of the sample and 8 μl of 6% Laemmli SDS reducing buffer (Alfar Aesar: J61337) was mixed and loaded into each well. HiMark Prestained protein standard (Thermo Fisher: LC5699) was used to monitor protein size. The samples were separated by 4–15% gradient Mini-PROTEAN TGX gel (Biorad). After transfer, the PVDF membrane was blocked with Biorad EveryBlot Blocking buffer for 5 mins. Next, the membrane was probed with mouse monoclonal RFP antibody [6G6] (chromotek, 1:1000 dilution) followed by HRP-conjugated Goat anti-mouse HRP-conjugated antibody (abcam: ab97255, 1:1000 dilution). Signals were detected using Supersignal West Pico chemiluminescent substrate (Thermo Scientific). The blot was stripped and probed with rabbit monoclonal beta-Actin (13E5) HRP-conjugated antibody (cell signaling, 1:1000 dilution). The signals were quantified using the Image lab software (Version 6.1.0).

**Statistical analyses.** Statistical analyses were carried out using Prism software (GraphPad) following established standards [100].

## Supporting information

**S1 Fig. Loss of other ADAM and ADAM-TS family members does not suppress *nekl* defects.** (A) Dot plot showing average brood sizes for 10 individual wild-type and *adm-2 (fd300)* mutant worms. (B) Bar plot showing the failure of most *C. elegans* ADAM family members to suppress molting defects in *nekl-2; nekl-3* mutants. (C) Table showing ADM-2 *C. elegans* orthologs and their corresponding human homologs. Error bars in A, B represent 95% confidence intervals. p-Values were determined using an unpaired t-test (A) (ns, $p \geq 0.05$) or Fisher's exact test (B): ****$p \leq 0.0001$. Raw data for this figure is provided in S1 File. (TIFF)

**S2 Fig. Alignment of *C. elegans* ADM-2 with human ADAMs.** Peptide alignment of *C. elegans* ADM-2 with human meltrin family members (ADAM9/12/19/33). Predicted domains of

ADM-2 are color coded. NLS, nuclear localization domain.
(PDF)

**S3 Fig. Additional images of ADM-2 expression.** (A–F and A'–F') Representative DIC (A–F) and confocal (A'–F') images of ADM-2 expression showing the anterior hypodermis (A, A'), nerve ring (B, B'), tail neurons (C, C'), and various stages of embryonic development (D–F'). Bar in A' = 10 μm (for A, A'); in B' = 10 μm (for B, B'); in C' = 10 μm (for C, C'); in F' = 10 μm (for D–F'). (G) Representative confocal image of an L2 larva expressing multi-copy P$_{adm-2}$::ADM-2::GFP in the plasma membrane of head neurons. Bar in G = 25 μm.
(TIFF)

**S4 Fig. Additional colocalization of ADM-2 with trafficking markers.** (A, B) Dot plots showing quantification of Mander's overlap coefficient for the overlap of ADM-2::mScarlet with GFP::CHC-1 and P$_{hyp7}$::HGRS-1::GFP proteins within the apical (A) and medial (B) planes. Mean values and 95% confidence intervals (error bars) are indicated. p-Values were calculated using an unpaired test: ****p ≤ 0.0001. (C–H) Representative confocal images of GFP::CHC-1 (C), P$_{hyp7}$::HGRS-1::GFP (F), and ADM-2::mScarlet (D, G) within the hyp7 medial plane. C'–H' are insets of C–H confocal images. Bar in E = 10 μm (for C–H); in E' = 5 μm (for insets C'–H'). (E',H') White arrows show ADM-2 large vesicular structures that do not colocalize with GFP::CHC-1 and P$_{hyp7}$HGRS-1::GFP puncta, which are lysosomes. Cyan arrows indicate vesicles containing ADM-2 that colocalize with GFP::CHC-1 and P$_{hyp7}$::HGRS-1::GFP. Raw data for this figure is provided in S1 File.
(TIFF)

**S5 Fig. ADM-2 and clathrin levels are increased upon weak loss of *mlt-3*.** (A,B) Dot plot showing the mean intensity (a.u.) of GFP::CHC-1 (A) and ADM-2::mScarlet (B) expression in the presence of Control RNAi (i.e., empty vector) and *mlt-3* RNAi. Group means along with 95% confidence intervals (error bars) are indicated. p-Values were obtained by comparing means using an unpaired t-test: **p ≤ 0.01, *p ≤ 0.05. Adult worms were imaged for this experiment. Raw data for this figure is provided in S1 File.
(TIFF)

**S6 Fig. Loss of dauer pathway function does not reverse *adm-2* suppression of *nekls*.** Bar plot showing suppression in *nekl-2; nekl-3 adm-2(fd130)* in the presence and absence of *daf-5* (*e1386*). Loss of DAF-5 leads to strong defects in the induction of the dauer pathway. p-Values were obtained by comparing means using Fisher's exact test. ns, p > 0.05. Raw data for this figure is provided in S1 File.
(TIFF)

**S7 Fig. Supplemental analysis of LRP-1 expression levels.** (A) Dot plot showing LRP-1::GFP mean intensity (a.u.) within the apical plane for each individual L4 worms in wild type and *adm-2(fd300)* backgrounds. (B) Dot plot showing LRP-1::GFP mean intensity (a.u.) within the apical plane for adults containing the P$_{hsp-16}$::*Empty vector* transgene in the absence of heat shock and after heat shock. Group means along with 95% confidence intervals (error bars) are indicated in plots A and B. p-Values were obtained by comparing means using an unpaired t-test: *p ≤ 0.05. (C) p-values obtained from comparisons of LRP-1::GFP mean intensity (a.u.) values in strains with the indicated transgenes after heat shock. ****p ≤ 0.0001; ***p ≤ 0.001, ns, not significant. (D) Florescence and corresponding DIC images of an adult worm carrying P$_{hsp-16}$::*adm-2*::*gfp* in the absence of heat shock and after heat shock. (E) Florescence and corresponding DIC images of an adult worm without the heat-shock array after heat shock. Scale bar in D = 50 μm (For all images in D). Scale bar in E = 20 μm (For images in E). Raw data for

this figure is provided in S1 File.
(TIFF)

**S8 Fig. Clathrin expression is not affected by loss of ADM-2 function.** (A,B) Representative confocal images of GFP::CHC-1 expression in the apical hyp7 region of the hypodermis in wild-type (A) and *adm-2(fd300)* null mutant (B) day-1 adult worms. Bar in A = 10 μm (for A, B). (C, D) Dot plots showing GFP::CHC-1 mean intensity (a.u.) (C) and the percentage of GFP-positive pixels (D) within the apical plane for individual worms of the specified genotype. In C and D, group means along with 95% confidence intervals (error bars) are indicated. p-values were obtained by comparing means using an unpaired t-test. ns, p > 0.05. Raw data for this figure is provided in S1 File.
(TIFF)

**S9 Fig. Functional CRISPR-based analysis of ADM-2 domains.** (A) Schematic representation of predicted protein domains within ADM-2. (B) Color-coded peptide sequence of ADM-2 corresponding to panel A. For additional details see S2 Fig. (C) Amino acid sequence change details of the ADM-2 variants generated using CRISPR methods. (D, E) Predicted three-dimensional protein structures (for amino acid region 307–328) of wild-type ADM-2 (orange) superimposed on modeled structures for CysL (D; violet), SH3-1(E; green) mutant proteins. Three conserved histidine residues (His312, His316, His322) of the Zn-metalloprotease domain are represented as sticks. A predicted 6.1-Å shift of the 'tele' nitrogen atom in the imidazole ring of His322 was predicted in the CysL variant.
(TIFF)

**S1 Table. List of all the strains used in this study.**
(PDF)

**S1 File. Compilation of raw data used in this study.**
(XLSX)

**S2 File. List of primers used in this study.**
(XLSX)

**S3 File. Detailed sequencing data for *adm-2* alleles.**
(PDF)

## Acknowledgments

We thank Amy Fluet for editing this manuscript and Barth D. Grant for the HGRS-1 marker strain.

## Author Contributions

**Conceptualization:** Braveen B. Joseph, David S. Fay.

**Data curation:** Braveen B. Joseph, Phillip T. Edeen, Sarina Meadows, Shaonil Binti, David S. Fay.

**Formal analysis:** Braveen B. Joseph, Phillip T. Edeen, Shaonil Binti, David S. Fay.

**Funding acquisition:** David S. Fay.

**Investigation:** Braveen B. Joseph, Phillip T. Edeen, Sarina Meadows, Shaonil Binti, David S. Fay.

**Methodology:** Braveen B. Joseph, Phillip T. Edeen, Sarina Meadows, Shaonil Binti, David S. Fay.

**Project administration:** David S. Fay.

**Resources:** David S. Fay.

**Supervision:** David S. Fay.

**Validation:** Braveen B. Joseph, Phillip T. Edeen, Sarina Meadows, Shaonil Binti, David S. Fay.

**Visualization:** Braveen B. Joseph, Sarina Meadows, David S. Fay.

**Writing – original draft:** Braveen B. Joseph, David S. Fay.

**Writing – review & editing:** Braveen B. Joseph, David S. Fay.

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
