## [Decision Letter · Decision Letter 0]

27 Jan 2022

Dear Dr Fay,

Thank you very much for submitting your Research Article entitled 'An unexpected role for the conserved ADAM-family metalloprotease ADM-2 in Caenorhabditis elegans molting' to PLOS Genetics.

The manuscript was fully evaluated at the editorial level and by independent peer reviewers. The reviewers appreciated the attention to an important topic but identified some concerns that we ask you address in a revised manuscript. Most comments likely can be addressed by rewriting, however reviewer 2 has made some suggestions for additional experiments that you may want to consider.

We therefore ask you to modify the manuscript according to the review recommendations. Your revisions should address the specific points made by each reviewer.

[LINK]

Yours sincerely,

Andrew D. Chisholm

Associate Editor

PLOS Genetics

Gregory P. Copenhaver

Editor-in-Chief

PLOS Genetics

Reviewer's Responses to Questions

**Comments to the Authors:**

Reviewer #1: This is a well written manuscript that describes fascinating work identifying the role of a conserved matrix metalloprotease, ADM-2, in roundworm molting, and provides evidence for a mechanism that includes endocytic trafficking that regulates ADM-2 activity and identify a candidate target, LRP-1, which may endocytose sterols that are used to create molting-specific steroid hormones that trigger molting.

While strong, the last piece of this puzzle, confirming their proposed target (LPR-1) is processed by their protease ADM-2, would strengthen support provided by their fluorescence imaging experiments for their model and, if possible, provide a satisfying conclusion. They have animals expressing the tagged target (LRP-1::GFP) in the relevant adm-2 backgrounds, and others (whom they cite) have shown by western that metalloproteases including ADM-2-like ADAM-12 cleaves the LRP-1-related low-density lipoprotein receptor-related protein.

Minor points:

“LGX” is generally written “LG X”.

S1 File: Is having more than one or two significant digits after the decimal (8 or 9 digits?) necessary and informative in the percent viability columns and the S7 Fig Average row?

S2 File:

The primer list sheet could use a little tidying up. SB3 doesn’t include sequence information, sometimes a forward primer is designated with “F” and sometimes “fwd”, and the capitalization of the sequences is inconsistent (if meaningful, what is the meaning?).

In the ADM-2 deletions tab, what is the heading for column C? Column C primers are listed by name (from the first sheet) but the screening and sequencing primers are not all named. Does it make sense to include all primers in the "Primer list" tab and include their use in a different column?

Reviewer #2: The authors have been analyzing endocytic vesicle trafficking in molting. In their previous work, they found nekl-2 and nekl-3 protein kinases play important roles in this process. In this manuscript, Joseph et al. identified novel mutations in adm-2 as the suppressors for the molting and larval arrest defects of nekl-2 and nekl-3 double mutants (both are weak alleles). Adm-2 encodes C. elegans ortholog of mammalian ADAM meltrin. ADM-2::GFP was broadly expressed and colocalized with endosomal and lysosomal markers in the hyp7 syncytium. The adm-2 suppressors failed to suppress the endocytic vesicle trafficking defects of nekl-2 and nekl-3 double mutants unlike their previously identified suppressor fcho-1. The authors found that the adm-2 affected the abundance of LRP-1 (sterol receptor) in the apical surface of hyp7; adm-2 loss-of-function increased LRP-1 levels, whereas adm-2 overexpression decreased them. Overall, this is an elegant genetic study analyzing the in vivo function of the ADAM family metalloprotease in animal development. Although all the experiments have been carefully conducted and the manuscript is well written, I think the presented data are still insufficient to support the authors’ model that “loss of adm-2 suppresses molting defects in nekl mutants by eliminating a negative regulator of LRP-1, thereby compensating for defects in the efficiency of LRP-1 and sterol uptake”.

Comments:

1. Based on their observation that the amount of LRP-1::GFP is upregulated in the adm-2(fd300) and is downregulated in adm-2 overexpression, the authors proposed a model that ADM-2 negatively regulates LRP-1: The extracellular domain of LRP-1 is released by direct or indirect action of the ADM-2 protease. However, I feel that the observed effects are quite weak, and it is unclear whether LRP-1 is actually cleaved and shed from the membrane. Because GFP is fused to the C-terminus (cytoplasmic domain?) of LRP-1, it is conceivable that GFP can be still anchored to the plasma membrane even after ectodomain shedding. If the authors wish to present their model, they should examine the proteolytic cleavage of LRP-1, for example by Western blotting experiments and compare the amounts of cleaved fragments between these two experimental conditions. They also need to examine the colocalization of ADM-2 and LRP-1.

2. It is not convincing that the adm-2 mutation promotes sterol uptake because more membrane-bound LRP-1 is available in the mutant background. I think LRP-1 should be endocytosed and transported to the lysosomes for sterol to be released to the cytosol. The authors, however, observed that apical accumulation of LRP-1 in nekl-3::AID failed to be suppressed by adm-2(fd318) (even slightly enhanced).

I feel it might be possible that ADM-2 may act independently of LRP-1. Although the authors consider that ADM-2 inhibits molting through its negative regulation of steroid hormone biosynthesis, another possibility is that ADM-2 may inhibit the process downstream of the NHR hormone receptors or a hormone-independent process. I think this possibility should be addressed for example by examining whether the lrp-1 null mutation can fully block the suppressing effect of adm-2 on nekl-2; nekl-3 defects. Also, it will be interesting to see if adm-2 can suppress the molting phenotype of the lrp-1 null mutant. If the authors observe any improvement in molting in adm-2; lrp-1 double mutants, that means that adm-2 can act in an lrp-1-independent pathway.

3. The ADM-2::GFP is expressed in various tissues including hypodermis. Although the authors believe that loss of the adm-2 function in hypodermis is responsible for the suppression, it is still unclear. The authors need to experimentally determine the responsible tissues for example by expressing adm-2 using tissue-specific promoters in the adm-2; nekle-2; nekl-3 mutant background or tissue-specific adm-2 RNAi experiments in the nekle-2; nekl-3 mutant background.

4. The patterns of expression of ADM-2 and LRP-1 were analyzed using mScarlet or GFP fusions. I think it is important to use functional fusion constructs to determine the protein localization. Please mention whether these constructs are functional or not.

5. It is reported that the cytoplasmic domain of human ADAM13 is cleaved and translocated into the nucleus to regulate neural crest cell migration and that the cytoplasmic domain of ADM-2 can be functionally replaced with that of ADAM13. Because mutations either in the catalytic domain or the cytoplasmic domain of ADM-2 can act as suppressors, I am curious if the authors observed the localization of ADM-2::GFP or ::mScarlet in the nucleus.

6. Although the authors isolated various alleles in adm-2, detail information for the mutation sites are provided for only part of the alleles. Please provide all the information if possible.

7. Fig. 4 M-O. Which focal plane do these photos depict?

8. Fig. S1. ADAM family members should be reworded as “ADAM and ADAMTS family members”. Also, in the text.

Minor comments:

1. Fig. 3C, D. Please indicate a scale bar.

2. line 200. ADM-2 can be adm-2.

3. line 226 S3D can be S3G.

4. line 235-236. ADM-2::GFP is not shown in Fig 4A or 4C.

5. line 265. ADM-2::mScarlet is not shown in Fig 5M-P.

6. line 316. ) can be missing.

7. line 553. Heat shock methods for Fig 6F and G are not mentioned.

Reviewer #3: Summary

This is a thoughtful and well-executed study of the mechanism of genetic suppression of nekl(rf) molting defects by the meltrin-family metalloprotease adm-2, revealing an interesting dynamic between negative and positive regulators of the molting process and clathrin-mediated endocytosis. The structure-function analysis of ADM-2 is impressively thorough. As described in greater detail below, the manuscript would benefit from:

1. inclusion of an fcho-1 control into panel 1G for comparison to adm-2 (and if possible, moving this panel to the start of figure 2).

2. examination of adm-2-dependent LRP-1:GFP expression during molting (in place of, or to complement, the adult data in Figure 6f-m)

3. use of fluorescence-only images (rather than DIC overlay) in Fig 6b,c and inclusion of a non-transgenic control

4. Minor text edits for clarity, described below.

Details

1. Figure 2 Examines whether adm-2 suppression of nekl-2/3 double-rf larval molting defects could be due to restoration of CME. Panel G from figure 1 would work better at the start of figure 2, and benefit from the inclusion of an fcho-1 control for comparison (eg. fcho-1; nekl-2). This would provide clearer physiological support for the LRP-1 localization data of figure 2, which differs in both the stage (adult vs. larvae) and the genetic background (NEKL-3 AID vs. nekl-2/3(rf) from the phenotype (molting defect suppression). A note on the choice of this system for examination of LRP-1 expression (eg., rather than performing nekl-3AID from hatching, examining at the onset of molting defects) would also be helpful to the reader.

2. Similarly, the examination of LRP-1:GFP expression in Figure 6 was performed in adults rather than larvae. I suggest adding an experiment examining adm-2-dependent changes in LRP-1 expression during molting, if possible. If LRP-1:GFP is below the limit of detection in larvae, it should be noted, with corresponding interpretations of adm-2 function expressed in more conservative language.

3. The heat shock dependence of the hs:ADM-4GFP transgene (Fig 6b,c) would be better illustrated by fluorescence-only images (black background) rather than fluorescence/DIC overlays, and by inclusion of a non-transgenic control.

4. Minor edits:

a. abstract: “Whereas loss of ADM-2 activity led to the upregulation of LRP-1, ADM-2 overexpression caused a reduction in LRP-1 abundance…” this sentence would benefit from an indication that this refers to the apical surface. eg) “Whereas loss of ADM-2 activity led to increased levels of apical LRP-1, ADM-2 overexpression caused a reduction in apical LRP-1 abundance…”

b. lines 45-46: the phrasing makes it sound like adm-2 regulates something that in turn negatively regulates LRP-1. Re-phrase for clarity?

c. line 101: remove the word ‘caused’ for clarity

d. line 114: move the word ‘specifically’ to just before the word de-repressing

e. For added transparency, the number of animals examined in 1E,F,G, 5c, 6a should be indicated.

f. line 126: change (Fig 1A and 1E) to just (Fig 1A), as there is no rescuing array in 1E and its description in line 128 is sufficient and appropriate.

g. line 136: change ‘The fd163 mutation’ to ‘The resulting intron retention’

h. line 160: replace the semi-colons with commas

i. line 178: replace the word ‘is’ with ‘may’ since this is based on lack of phenotypes from RNAi

j. line 233-234: the words ‘rare’ and ‘occasionally’ are redundant with each other

k. lines 237-239: the faintness of ADM-2 expression in hyp7 is postulated to be due to rapid turnover, possibly mediated by CME. This would make more sense to the reader if it were noted that detection at the plasma membrane was difficult to detect 'relative to other compartments'.

l. line 320: the words ‘specifically’ and ‘only’ are redundant with each other

m. Figure 6 title could be more informative. Instead of “and LRP-1”, maybe “and apical abundance of LRP-1”?

n. The font under the scale bars in panels 6C and E should be increased, or omitted altogether and described only in the figure legend, as is the case for panel D.

o. In Figure 6m, the Phsp-16::adm-2+ genotype is shortened to Padm-2+, but hs:adm-2+ or similar might be clearer.

**Have all data underlying the figures and results presented in the manuscript been provided?**

Reviewer #1: Yes

Reviewer #2: Yes

Reviewer #3: Yes

PLOS authors have the option to publish the peer review history of their article (what does this mean?). If published, this will include your full peer review and any attached files.

Reviewer #1: **Yes: **Tina L. Gumienny

Reviewer #2: No

Reviewer #3: No

---

## [Decision Letter · Decision Letter 1]

11 May 2022

Dear Dr Fay,

We are pleased to inform you that your manuscript entitled "An unexpected role for the conserved ADAM-family metalloprotease ADM-2 in Caenorhabditis elegans molting" has been editorially accepted for publication in PLOS Genetics. Congratulations!

Yours sincerely,

Andrew D. Chisholm

Associate Editor

PLOS Genetics

Gregory P. Copenhaver

Editor-in-Chief

PLOS Genetics

Comments from the reviewers (if applicable):

Reviewer's Responses to Questions

**Comments to the Authors:**

Reviewer #2: The authors mostly addressed my concerns. Their new finding in western blot analysis that ADM-2 regulates apical LRP-1 levels independently of its catalytic activity is interesting. Although the mechanism underlying their observation is unclear, it can provide an important clue to understand the action of ADAM proteases in LRP regulation and signal transduction “in vivo”.

The authors also found that (RNAi) knockdown of adm-2 did NOT suppress molting defects in an lrp-1 null mutant, which could be interpreted as consistent with ADM-2 acting through LRP-1. (We note that these RNAi conditions were identical to those in which we observed nekl-2; nekl-3 suppression.)

Although they do not mention this finding in their manuscript, I think the data would be better to be included to indicate the functional significance of ADM-2 regulation of LRP-1 levels.

Line 353: “this” is repeated.

Reviewer #3: This revision brings to light an interesting complexity in adm-2 function. While the suppression of the nekl molting defect is dependent on metalloprotease (MTP) activity (as well as other domains), adm-2 regulation of LRP-1 does not appear to involve proteolytic processing, as evidenced by Western blots of WT vs. adm-2 null mutants and by examination of LRP-1 levels in adm-2 MTP-defective mutants as well as in hs:adm-2MTP(–) animals. In addition, based on previous observations implicating dauther pathway function in the regulation of molting, the authors add data examining whether daf-5(lf) impacts the ability of adm-2 to suppress the nekl molting defect and they find that it does not.

While the main targets of adm-2 in the suppression of the nekl molting defects are yet to be identified, this analysis provides insight into what promises to be a multifunctional member of the ADAM family. The authors have addressed my concerns with their original submission, and I have no additional changes to request.

**Have all data underlying the figures and results presented in the manuscript been provided?**

Reviewer #2: Yes

Reviewer #3: Yes

PLOS authors have the option to publish the peer review history of their article (what does this mean?). If published, this will include your full peer review and any attached files.

Reviewer #2: No

Reviewer #3: No

**Data Deposition**

http://datadryad.org/submit?journalID=pgenetics&manu=PGENETICS-D-21-01647R1

**Press Queries**

---

## [Editor Report · Acceptance letter]

25 May 2022

PGENETICS-D-21-01647R1 

An unexpected role for the conserved ADAM-family metalloprotease ADM-2 in Caenorhabditis elegans molting 

Dear Dr Fay, 

We are pleased to inform you that your manuscript entitled "An unexpected role for the conserved ADAM-family metalloprotease ADM-2 in Caenorhabditis elegans molting" has been formally accepted for publication in PLOS Genetics! Your manuscript is now with our production department and you will be notified of the publication date in due course.

With kind regards,

Livia Horvath

PLOS Genetics

On behalf of:
